# Mapping out Min protein patterns in fully confined fluidic chambers

Yaron Caspi, Cees Dekker*

Department of Bionanoscience, Kavli Institute of Nanoscience, Delft University of Technology, Delft, Netherlands

**Abstract** The bacterial Min protein system provides a major model system for studying reaction-diffusion processes in biology. Here we present the first *in vitro* study of the Min system in fully confined three-dimensional chambers that are lithography-defined, lipid-bilayer coated and isolated through pressure valves. We identify three typical dynamical behaviors that occur dependent on the geometrical chamber parameters: pole-to-pole oscillations, spiral rotations, and traveling waves. We establish the geometrical selection rules and show that, surprisingly, Min-protein spiral rotations govern the larger part of the geometrical phase diagram. Confinement as well as an elevated temperature reduce the characteristic wavelength of the Min patterns, although even for confined chambers with a bacterial-level viscosity, the patterns retain a ~5 times larger wavelength than *in vivo*. Our results provide an essential experimental base for modeling of intracellular Min gradients in bacterial cell division as well as, more generally, for understanding pattern formation in reaction-diffusion systems.

## Introduction

The Min protein system determines the localization of the division site in a wide range of bacterial cells (*Loose et al., 2011b*; *Lutkenhaus, 2012*; *Shih and Zheng, 2013*; *Rowlett and Margolin, 2015*). In *Escherichia coli* (*E. coli*), Min proteins dynamically oscillate from pole-to-pole on a typical time scale of about 1 min (*Raskin and de Boer, 1999*; *Hu and Lutkenhaus, 1999*; *Hale et al., 2001*). Reaction-diffusion mechanisms were invoked in order to explain these oscillations (*Kruse et al., 2007*), and as of today, the Min system is one of the most prominent examples of intracellular pattern formation in biology (*Soh et al., 2010*).

Three proteins participate in the *E. coli* Min oscillations: (i) MinD, an ATPase that can bind the plasma membrane through a short amphipathic peptide (*Hu and Lutkenhaus, 2003*; *Szeto et al., 2003*) in a cooperative manner (*Mileykovskaya et al., 2003*; *Renner and Weibel, 2012*). Though it was commonly assumed that it binds the membrane only in its ATP-bound form (*de Boer et al., 1991*; *Hu and Lutkenhaus, 2001*; *Lackner et al., 2003*), a recent work showed that MinD can also bind the membrane in the ADP-bound form (*Zheng et al., 2014*). (ii) MinE, a protein that is recruited to the membrane by MinD (*Ma et al., 2003*), upon which it induces MinD's ATPase activity causing MinD to be released from the membrane (*Hu et al., 2002*). Subsequently, while diffusing in the cytosol, an exchange of ADP to ATP occurs, and the MinD proteins re-enter the cycle by rebinding the membrane. Several authors showed that MinE can persist on the membrane after MinD detachment (*Loose et al., 2011a*; *Park et al., 2011*) or can even interact with the membrane by itself (*Hsieh et al., 2010*; *Shih et al., 2011*; *Zheng et al., 2014*). The exact contribution of this process to the overall Min dynamics remains unclear. Finally, (iii) MinC protein is also recruited to the membrane by MinD, but, due to overlap binding with MinE, is released from MinD after MinE binding (*Hu et al., 2003*; *Ma et al., 2004*; *Wu et al., 2011*). While MinC is the sole member of the system that directly interacts with the division apparatus (*Hu et al., 1999*; *Cordell et al., 2001*;

*For correspondence: C.Dekker@tudelft.nl

**Competing interests:** The authors declare that no competing interests exist.

**eLife digest** Some proteins can spontaneously organize themselves into ordered patterns within living cells. One widely studied pattern is made in a rod-shaped bacterium called *Escherichia coli* by a group of proteins called the Min proteins. The pattern formed by the Min proteins allows an *E. coli* cell to produce two equally sized daughter cells when it divides by ensuring that the division machinery correctly assembles at the center of the parent cell. These proteins move back and forth between the two ends of the parent cell so that the levels of Min proteins are highest at the ends and lowest in the middle. Since the Min proteins act to inhibit the assembly of the cell division machinery, this machinery only assembles in locations where the level of Min proteins is at its lowest, that is, at the middle of the cell.

When Min proteins are purified and placed within an artificial compartment that contains a source of chemical energy and is covered by a membrane similar to the membranes that surround cells, they spontaneously form traveling waves on top of the membrane in many directions along to surface. It is not clear how these waves relate to the oscillations seen in *E. coli*. Caspi and Dekker now analyze the behavior of purified Min proteins inside chambers of various sizes that are fully enclosed by a membrane.

The results show that in narrow chambers, Min proteins move back and forth (i.e. oscillate) from one side to the other. However, in wider containers the wave motion is more common. In containers of medium width the Min proteins rotate in a spiral fashion. Caspi and Dekker propose that the spiral rotations are the underlying pattern formed by Min proteins and that the back and forth motion is caused by spirals being cut short. In other words, if a spiral cannot form because the compartment is too small then the back and forth motion emerges. Similarly, Caspi and Dekker propose that the waves emerge in larger containers when multiple spirals come together.

These findings suggest that the different patterns that Min proteins form in bacterial cells and artificial compartments arise from different underlying mechanisms. The next step will be to investigate other differences in how the patterns of Min proteins form in *E. coli* and in artificial compartments.

*Dajkovic et al., 2008*), it is believed to be only a passive hitchhiker that does not determine the dynamical behavior of the system. Thus, only MinD and MinE are needed in order to form dynamical pole-to-pole oscillations in *E. coli* cells.

A number of relevant properties of the Min system were identified when *E. coli* bacteria were perturbed from their native rod-shape form. When cells were grown as filaments, a dynamic series of Min bands with a characteristic length of ~8 µm was observed (*Raskin and de Boer, 1999*). When the filamentous cells were grown at different temperatures, this length scale did not change but the temporal period of the oscillations decreased according to an Arrhenius law (*Touhami et al., 2006*). Likewise, in oval-shape cells, the Min system preferentially oscillated along the longest axis (*Corbin et al., 2002*; *Shih et al., 2005*). Interestingly, while in round ΔMreB cells, the Min oscillation occurred, in the majority of the cases, from one end of the cell to the other in a well-defined manner, in rounded rodA-amber-mutation cells, the oscillation direction moved chaotically from one spot along the membrane to another. A similar mode of chaotic oscillations was also observed when cells adopted aberrant shapes upon getting squeezed into slits smaller than their natural width (*Männik et al., 2012*). In addition, when *E. coli* cells were mutated to form a Y shape, a sequence of oscillation nodes along the cell arms was observed that depended on the relative length of the Y shape arms (*Varma et al., 2008*). These results show that the Min system can adapt to the cell geometry and modify its dynamical behavior accordingly. Indeed, in a recent example, Wu et al. sculpted *E. coli* cells into various rectangular and square shapes. They found that the Min system behavior was characterized by a typical length scale of 3–6 µm. In addition, they showed that the choice for a particular Min pattern was governed by the symmetries of the cell shape, its aspect ratio, and its size. Consequentially, a variety of Min patterns were observed, including rotational, transversal, and longitudinal modes (*Wu et al., 2015*).

In parallel with these cell-biology studies, *in vitro* reconstitution of the Min system on supported lipid membranes (SLB) significantly advanced the understanding of its dynamical mechanism (*Loose et al., 2008*). On two-dimensional (2D) SLBs that are much larger than the typical length scale of the Min patterns, instead of oscillations, the Min proteins exhibit patterns that can be grouped into two classes: rotating spirals and traveling waves. In general, traveling waves were formed when bands that emanated from two counter-rotating spirals collided. Importantly, the wavelength of the traveling waves was very large, between 65–100 μm, and depended on the ratio of MinD to MinE concentration. Thus, when the Min system is reconstituted *in vitro*, it exhibits a pattern-formation behavior with a typical spatial dimension that is about an order of magnitude larger than the one observed *in vivo*. Interestingly, when the Min system was reconstituted under limiting concentration conditions, unique patterns, in particular a bursting type, were observed (*Vecchiarelli et al., 2016*). It was suggested that these patterns might be more closely related to the behavior of the Min system *in vivo*.

One of the intriguing properties of the *in vitro* behavior of the Min system is its ability to adapt to geometrical patterns that are embedded in the SLB. For example, when rectangular patches of flat surface SLB were separated one from the other with gold barriers that were much larger than the characteristic size of the Min dynamics *in vitro* (~100 μm), Min waves propagated in a direction that depended on the aspect ratio of the patch (*Schweizer et al., 2012*). Similarly, Min waves can be oriented within an SLB if parallel grooves are molded into the surface (*Zieske et al., 2014*). However, the geometry-selection rules that were found in these *in vitro* experiments do not correspond to the ones that were observed *in vivo* (*Wu et al., 2015*). Recently, utilizing partly confined fabricated reaction chambers, Zieske and Schwille were able to reproduce Min pole-to-pole oscillations as well as double and triple-band standing waves, similar to the patterns observed in filamentous *E. coli* (*Zieske and Schwille, 2013*, *2014*). Similarly, Zieske et al. showed that when grooves had a shape similar to that of dividing bacteria at the last division stage, the Min proteins stochastically distributed between the two sides of the grooves. This observation is similar to the way that Min proteins are distributed between the two progenies of an *E. coli* mother cell (*Di Ventura and Sourjik, 2011*). However, these *in vitro* Min oscillations were stable only when the groove width was much smaller than the *in vitro* wavelength of the waves. Since the geometrical selection rules of the Min patterns that were established in sculpted cells (*Wu et al., 2015*) do not coincide with the ones that were established for the *in vitro* grooves, the exact relation between the underlying mechanisms of these two phenomena of Min proteins remains unclear.

Recalling that the Min dynamics is a reaction-diffusion process, both the reaction and the diffusion parameters may control the behavior of the system *in vitro* as well as *in vivo*. Several experimental attempts have been made to study these reaction-diffusion factors. For example, when the overall bulk diffusivity of the proteins was decreased by a factor of 10, the Min wavelength was, surprisingly, only marginally reduced (*Schweizer et al., 2012*). In contrast, when the Min dynamics was studied on the outer side of giant unilamellar vesicles (GUVs), both the wavelength and wave velocity increased considerably relative to the behavior on an SLB (*Martos et al., 2013*). It was suggested that the 4-fold increase in the diffusion rate of the Min protein on the GUVs membrane was responsible for this phenomenon. Concerning the reaction rates, by increasing the salt concentration or the ratio of anionic to natural phospholipids, Vecchiarelli et al. observed a reduction in the size of the Min bands (*Vecchiarelli et al., 2014*), which they attributed to the different affinity of MinE to the membrane under these conditions. Similarly, Zieske and Schwille observed a decreased wavelength to a value of about 25 μm by increasing the concentration of a negatively charged lipid, cardiolipin, in the SLB to an (unphysiologically high) molar ratio of 70% (*Zieske and Schwille, 2014*). Note that even this reduced value is still much larger than the characteristic ~5 μm size scale measured *in vivo*.

Over the years, many mathematical models have been constructed for the Min system (*Meinhardt and de Boer, 2001*; *Huang and Wingreen, 2004*; *Meacci and Kruse, 2005*; *Fischer-Friedrich et al., 2007*; *Arjunan and Tomita, 2010*; *Hoffmann and Schwarz, 2014*). Each of them postulated a somewhat different molecular mechanism and was able to reproduce several of the observed Min phenomena for certain values of the model's parameters. However, the ability to explain all Min behaviors and the robustness to changes in the parameters was less well considered. In particular, the ability to gap the *in vitro* and *in vivo* behavior of the system was not given much attention. Recently, Bonny et al. claimed that they were able to bridge this gulf (*Bonny et al., 2013*). However, they were only able to reproduce the *in vivo* oscillations and *in vitro* surface waves for

reaction parameters that were different in both cases over several orders of magnitudes, and the experimental basis for this variation remained unclear. To date, the best Min model was developed by Halatek and Frey, based on a previous model by Huang and Wingreen (*Huang et al., 2003*; *Halatek and Frey, 2012*; *Thalmeier et al., 2016*). This model is able to reproduce most of the *in vivo* experimental results over a wide range of geometries and conditions (*Wu et al., 2015*). The ability of this model to reproduce the Min behavior *in vitro* was, however, not yet reported.

One reason for the lack of theoretical ability to reproduce all observed Min behaviors using a single set of parameters is the lack of comprehensive experimental results for the *in vitro* behavior of

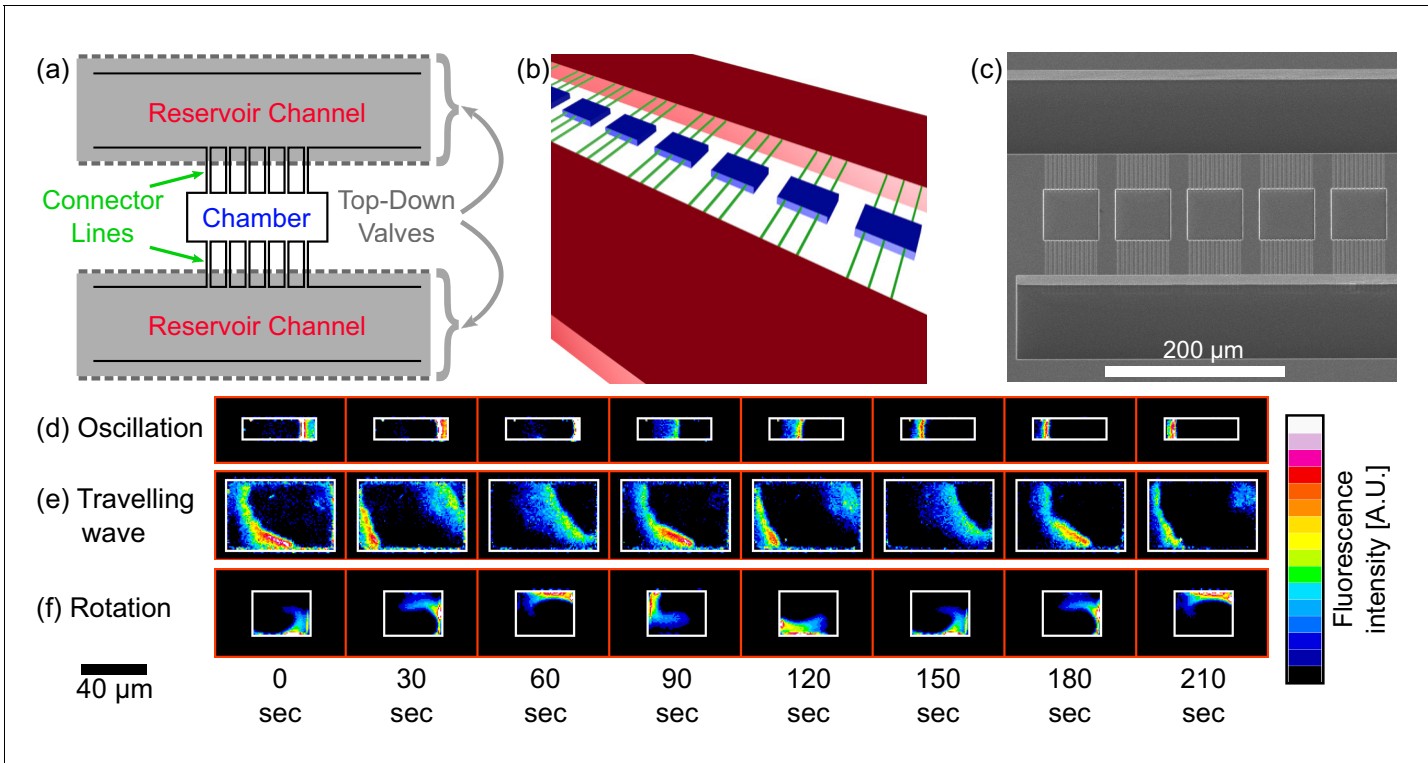

**Figure 1.** Chip structure and basic modes of Min patterns. (a) Top view illustration of the microfluidic device structure. We study Min protein pattern formation in totally enclosed microfluidic chambers. Each chamber (blue) is 2.4 μm high with a width ranging from 10 to 60 μm and a length ranging from 10 to 90 μm. Each chamber is connected through small connector lines (green, with a cross-section of ~0.9 × 0.9 μm) from both sides to two ~30 μm deep reservoirs channels (red). A top layer of pressurized microfluidic valves (gray shading) are placed above the reservoirs channels. After the walls of the chambers are coated with a lipid bilayer, MinD and MinE proteins are injected into the device. Subsequently, the valves above the reservoir channels are closed. Consequently, the rectangular chambers (including the connector lines) are separated from the rest of the device. This unique structure allows studying the 3D geometric selection rules of the Min system *in vitro*. (b) 3D illustration of the microfluidic device structure. Chambers are in blue, connector lines in green, and reservoir channels in red. For clarity, the pressure valves are not shown. (c) SEM image of the silicon wafer master that was used in order to replicate the chambers, connector lines and reservoir channels into PDMS. Note that the fabrication was done with a positive resist and, thus, we have used a double replica method in order to recover the right orientation of the chambers (see Materials and methods). (d–f) Characterization of dynamical Min patterns observed in the confined fluidic chambers. (d) Oscillations - Min proteins periodically move back and forth between two poles of the chamber. (e) Traveling waves - Min proteins wave fronts continuously propagate from one side of the chamber to the other. (f) Rotations - Min zone circulates around a fixed point in the chamber. Scale bar 40 μm applied to all three examples (d–f). Fluorescence signals represent MinE patterns.

The following figure supplements are available for figure 1:

**Figure supplement 1.** Schematics of chip fabrication and operation.

**Figure supplement 2.** SEM images of two different areas in the lower layer PDMS chip.

**Figure supplement 3.** Schematic representation of the overall lower layer chip structure.

Min proteins under a wide range of geometrical confinements. First steps in this direction were taken by Zieske and Schwille (*Zieske and Schwille, 2013*, *2014*). However, their compartment were only semi-confined and they probed the system behavior mainly for compartments that were much narrower than the typical Min wavelength, and hence, a full description of the geometrical selection rules for the Min system in confined chambers, and especially the relation between *in vitro* oscillations and *in vitro* traveling waves, is still missing.

Here, we studied the pattern formation behavior of MinD and MinE proteins inside fully 3D confined chambers, under the aim to better understand the relations between different factors that determine the influence of geometry on the dynamics of the Min system *in vitro*. To obtained these chambers and encapsulate Min proteins inside them, we fabricated PDMS chips and coated all the chamber walls with SLB before injecting the Min proteins and spatially isolating the chambers from the rest of the chip through soft-lithography PDMS valves. We determine the Min patterns as a function of the geometrical factors of the chambers as well as of other factors such as temperature and viscosity, and compared these results to the one that are observed with flat SLB. We show that the rotational Min spirals can evolve to pole-to-pole oscillations if the dimensions of the confinement are reduced or to traveling waves if the dimensions of the confinement are increased. We provide a phase diagram that maps out the Min patterns over a wide range of geometrical parameters. In addition, we show that several parameters, including 3D confinement, medium viscosity and temperature can reduce the wavelength of traveling waves. Yet, all these parameters did not resolve the *in vitro*/*in vivo* dichotomy. Our comprehensive set of data, however, provides essential information that is needed in order to understand the molecular mechanism of the Min system, which underlies its essential pattern-formation abilities.

## Results

### Fabrication and preparation of spatially confined fluidic microchambers

We fabricated microfluidic chips that are composed of two stacked PDMS layers (see *Figure 1a–c* and *Figure 1—figure supplement 1* to *Figure 1—figure supplement 3*). The bottom layer had structures of three different heights: (i) Rectangular chambers with height of ~2.4 μm and lateral dimensions ranging from 10 × 10 μm to 60 × 80 μm (blue part in *Figure 1b*); (ii) Reservoir channels with height of ~30 μm and width of 100 μm (red part in *Figure 1b*); and (iii) thin connectors channels with height of ~0.9 μm and width of ~0.9 μm (green part in *Figure 1c*) that connect the reservoirs and the chambers. The upper layer of the device consisted of air-filled PDMS channels that were aligned directly above the reservoir channels and connected to a high-pressure argon line thus serving as pneumatic pressure valves (*Unger et al., 2000*). Upon increasing the pressure of the air-filled channels in the upper layer, the ceiling of the lower layer (thickness ~20 μm) deflected downward above the reservoir channels and closed the entrance to the connector lines. Using this design, we were able to spatially isolate the central PDMS chambers (blue in *Figure 1b*) from the rest of the chip

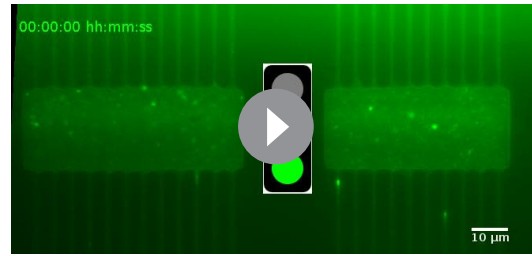

**Video 1.** Isolation of 3D confined chambers from the rest of the chip. After the SLB was formed in the device, as described in the Materials and methods, the chip was connected to a syringe pump containing Min buffer. A flow of 75−150 μl/h was applied (green traffic light). The application of buffer flow resulted in migration of the SUVs that did not splash during the previous incubation period on the chambers walls along the stream lines. After ~3 min the valves were closed and the flow was stopped (red traffic light). This resulted in an immediate halt of the SUVs flow, showing that the chambers were isolated from the rest of the chip. In order to show reversibility of the valves operation, after additional ~1.5 min, the valves were opened (second round of green traffic light). Immediately upon releasing the pressure in the pressure valves, the SUVs started to flow again, showing that the halting of the flow was the result of the valves operation and did not result from, say, stopping the syringe pump operation. Of course, stopping the operation of the syringe pump will also result in a halt of the flow in the microfluidic device. However, due to pressure-difference equilibration, this process usually lasts couple of minutes. In contrast, upon operating the valves, an immediate halt in the flow was observed.

and to obtain totally isolated 3D confined volumes for studying the Min patterns.

We used the traditional method of rupture and spreading of small unilamellar vesicles (SUVs) for the formation of an SLB onto all inner surfaces within our microfluidic device (SUVs composition 67% DOPC, 33% DOPG, supplemented with 0.03% TopFluor cardiolipin for fluorescent imaging) (*Brian and McConnell, 1984*; *Richter et al., 2006*). After the device was flushed with SUVs and incubated for ~1 hr, a nice continuous SLB was formed on all the inner walls of the chambers (data not shown). This feature of our microchambers that are covered by an SLB from all sides is what distinguishes our approach from previous studies that reconstituted Min proteins in fabricated structures (*Loose et al., 2008*; *Ivanov and Mizuuchi, 2010*; *Schweizer et al., 2012*; *Martos et al., 2015*; *Zieske and Schwille, 2013*; *Vecchiarelli et al., 2014*; *Zieske and Schwille, 2014*; *Vecchiarelli et al., 2016*). Subsequently, the device was extensively washed in order to remove the residual SUVs. For several devices, we checked the formation of a fluid SLB using fluorescence recovery after photobleaching (data not shown); for other devices, we relied on the homogeneous fluorescence signal of the TopFluor cardiolipin. It was important, for each device, to check that the valves worked properly. As a quality control, we checked the functionality of the valves by operating them during the first stages of the washing process and observing the effect on the flow through the microchambers (see *Video 1* and the Materials and methods section). These results corroborated that upon closing the pressure valves, we obtained truly confined chambers in our microfluidic devices.

After the device was washed and the valves were tested for functionality, the chambers were filled with a solution containing MinD and MinE that we purified and labeled beforehand (MinE 0.8 µM: MinE-Cy5 0.2 µM: MinD 0.9 µM: MinD-Cy3 0.2 µM), supplemented by ATP (5 mM) and an ATP-regeneration system (see Materials and methods). Subsequently, we closed the pressure valves and, after an incubation period of >1 hr, we recorded movies of the resulting dynamical behavior of the Min system.

## A variety of Min patterns in the microchambers

We observed a rich set of dynamical phenomena for the spatio-temporal behavior of the Min proteins, and studied it in hundreds of confined chambers. In most of the cases, the Min proteins showed a defined pattern in each chamber that was stable for the duration of the movie acquisition (~10 min). Three primary dynamical behaviors stood out as the most prominent patterns; (i) pole-to-pole oscillations - where the Min proteins periodically move back and forth from one side of a structure to the other (see *Figure 1d*); (ii) traveling waves - where the proteins move constantly from one side of a structure to the opposite side (see *Figure 1e*); and (iii) spiral rotations - where the proteins constantly rotate in a spiral fashion within the microchamber (see *Figure 1f*). As can be seen from *Figure 1d–f*, the typical time it took to reestablish the Min zone in these microchambers was 1−5 min.

For each chamber, we recorded a movie with the dynamical behavior of the Min system. To represent the temporal behavior in a specific chamber in the format of a still image, we adopted a quadrant scheme, see *Figure 2a*. Each quadrant is composed of: (top-left) a single frame from a movie of the dynamical pattern in this chamber; (bottom-left) an X-axis kymograph of the Min concentration along the horizontal chamber middle (red line in top-left image of *Figure 2a*); (top-right) a Y-axis kymograph of the Min concentration along the vertical chamber middle (blue line in top-left image of *Figure 2a*); and (lower-right) a temporal-standard-deviation (STD) image of the movie. In this scheme, each basic type of dynamical behavior such as oscillations, waves or spiral rotations has a distinct signature.

For example, for side-to-side oscillations (cf. *Figure 2b*), the top-left representation of the quadrant may show a snapshot of the Min concentration at one side of the chamber. The Xt kymograph shows the movement of the Min proteins along the X-axis as they travel from the chamber middle to one side, then to the chamber middle, then to the other side of the chamber, and so on. This repetitive reappearing of Min proteins at the chamber middle results in oblique lines in the Xt kymograph that run from the Kymograph middle to both of its sides. In contrast, the reappearing of Min proteins in the middle of the chamber results in repetitive vertical lines in the Yt kymograph. Finally, the STD image shows the locations where the variations in the Min proteins are the largest, i.e., at the chamber sides in this example.

For a traveling wave (cf. *Figure 2c*), the top-left single frame image in the quadrant representation can show, for example, two wave fronts propagating from the bottom right corner of the

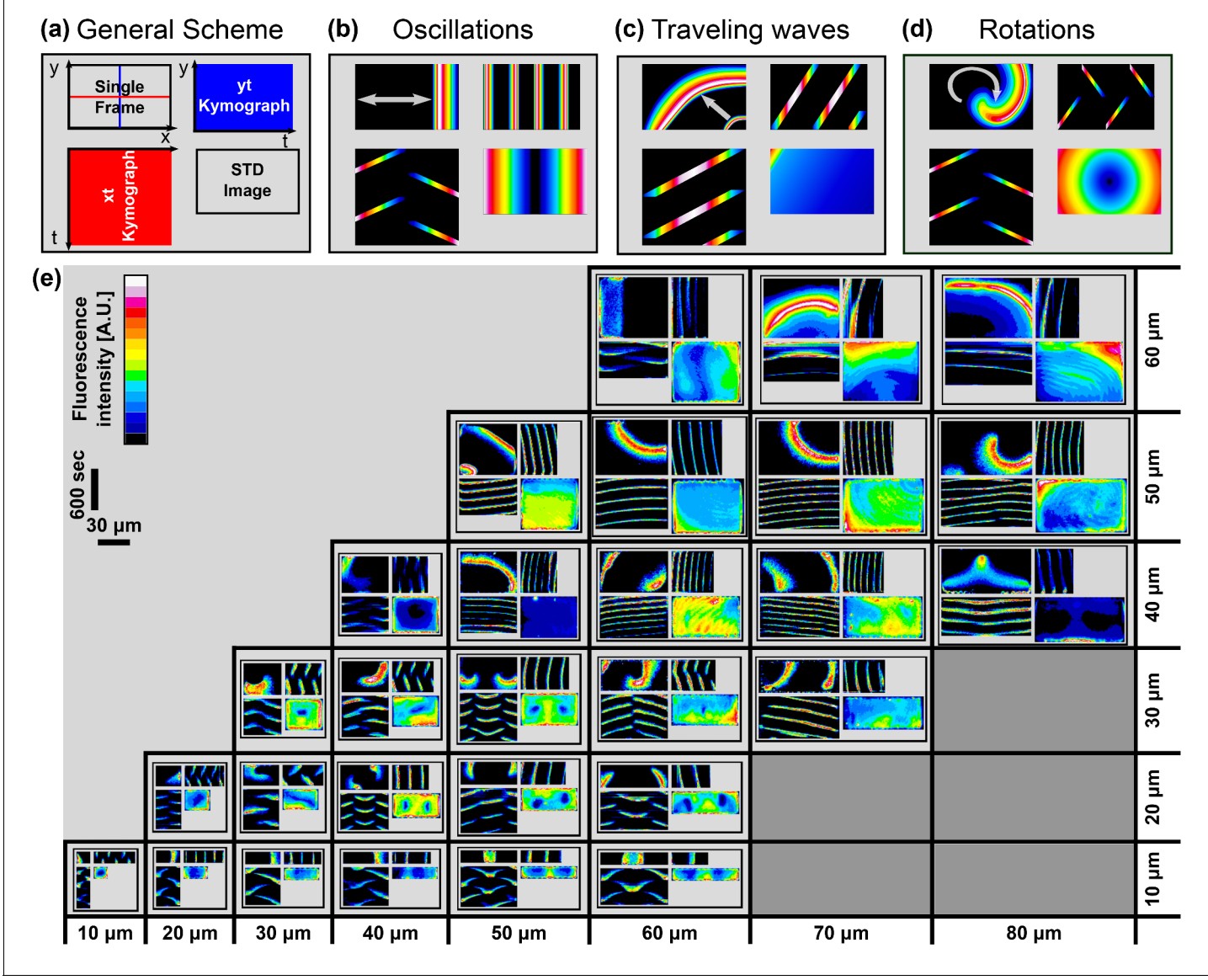

**Figure 2.** Atlas of the Min system behavior in 3D confined chambers. We represent the dynamical Min behavior in each case as a quadrant. (a) Illustration of the quadrant scheme. Each quadrant is composed of (upper left) A single frame from a movie that captures the behavior of the Min system in a specific chamber at a specific point of time; (lower right) A temporal standard deviation picture of the movie; (lower left) an Xt-kymograph of the middle line cross-section of the chamber along the red line in the upper left movie frame image, and (upper right) a Yt-kymograph of a middle line cross-section of the chamber along the blue line in the upper left movie frame image. (b–d) Illustrative examples of the three pure genera of Min dynamics that were observed. (b) Oscillations. (c) Traveling waves, and (d) Rotations. (e) Table representing real examples of observed Min dynamics in the chambers, organized according to the quadrant scheme. Each image is color coded in a 16-colors look-up table as shown in the legend. Scale bar represents 30 μm in the x and y directions and 600 s for the *Xt* and *Yt* kymographs. A detailed explanation for the quadrant representation of the Min pure and non-pure behaviors is found in the main text. Fluorescence signals represent MinE patterns.

The following figure supplement is available for figure 2:

**Figure supplement 1.** A zoom-in of the atlas of the Min system behavior in 3D confined chambers.

chamber to the opposite corner. The Xt and Yt kymographs for this case show a typical signature of continuous oblique lines that are formed as the Min proteins pass through the mid-lines of the chamber. The STD figure merits a comment. For an infinite-time movie of fully homogeneous waves, the

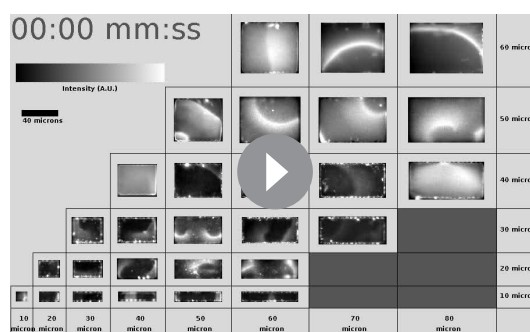

**Video 2.** Movies of Min patterns formation in chambers with different sizes that were used for the construction of *Figure 2e* of the main text.

STD figure of a traveling wave should have a uniform flat profile. For a finite-time recording of a wave propagation, the STD figure will, however, show a slightly monotonically decreasing profile.

For the rotational behavior (cf. *Figure 2d*), the repetitive passage of the Min proteins along both the X-axis and the Y-axis chamber midlines results in oblique lines that run from the kymograph middle to its sides, for both the Xt and Yt kymographs. The STD image of Min protein rotation shows the focal point and a symmetric concentric gradient profile around it.

*Figure 2e* represents a typical repertoire of the Min dynamical behaviors for the different geometries of the chambers. A corresponding composite *Video 2* is also provided (very worthwhile to examine this movie, as it illustrates the intrinsic dynamic patterns particularly well). In addition, a zoom-in for the smaller chambers sizes is shown in *Figure 2—figure supplement 1*. In this atlas of dynamical patterns, one can observe a variety of patterns, e.g., side-to-side oscillations in the 30 × 10 μm chamber, a traveling wave in the 60 × 50 μm chamber, and a spiral rotation in the 40 × 40 μm chamber. More complex dynamical patterns were also observed. For example, in chambers of size 50 × 10 μm and 60 × 10 μm, instead of the most simple side-to-side oscillation, we observed a striped pattern, where the Min concentration oscillated back and forth from the center of the chamber to both sides (see *Figure 2—figure supplement 1*). This is similar to what has been observed in filamentous cells (*Raskin and de Boer, 1999*; *Touhami et al., 2006*), shaped long cells (*Wu et al., 2015*), and *in vitro* grooves (*Zieske and Schwille, 2014*). Furthermore, in some cases we observed more than one rotational center within a single microchamber, a behavior that can be seen in the 40 × 20 μm, 50 × 20 μm, 60 × 20 μm and 50 × 30 μm chambers.

## Analysis of the Min pattern formation in confined microchambers

We set out to quantify the relation between the dimensions of the chamber and the preferred dynamical behavior of the Min system. Altogether we analyzed the dynamical behavior of the Min proteins in 553 different chambers. For each chamber that was recorded, we identified to what class of dynamical behaviors the observed pattern conform to: (i) oscillations. (ii) traveling waves, or (iii) rotations. Cases with more than one rotational center in the chamber where tagged as rotations and cases of striped or side-to-side oscillations where tagged as oscillations. The large majority of observed patterns (>95%) could be readily classified in these three categories. Naturally, there exist some borderline cases between the three types. For example, we observed that a rotational center of the Min proteins could propagate from one side of the chamber to the other. This behavior was grouped under the tag of traveling waves.

We thus analyzed the preferred dynamical behavior of the Min system in the chambers in terms of the geometrical parameters of the chambers such as the width (*W*) length (*L*), aspect ratio (*L/W*), and area (*L × W*) of the chambers. The results are shown in *Figure 3a–c* and *Figure 3—figure supplement 1a*. As can be seen, clear relations exist between the geometry of the chamber and the observed dynamical Min behavior. We note a few particular features: (i) Rotational patterns appear as the majority for chambers with small aspect ratio (*Figure 3a*). As the chamber aspect ratio increases, the probability to obtain rotational behavior decreases, and for chambers with an aspect ratio larger than 2.8, we hardly observed rotational behavior in our chambers. (ii) Oscillatory behavior predominantly appears when the chamber width is small (~10 μm) (*Figure 3b*). (iii) Traveling waves mainly appear if the chamber length is relatively large (*Figure 3c*), and their prevalence increases as the chamber length increases. Similarly, if the chamber area is relatively large the prevalence of obtaining a traveling waves increases (see. *Figure 3—figure supplement 1a*). For these large areas, the confined chambers reflect the surface waves that were observed on unbound SLBs (*Loose et al., 2008*; *Vecchiarelli et al., 2014*). The clear relation found between the chamber geometry and the dynamical Min patterns unambiguously proves that confinement sets the rules for Min

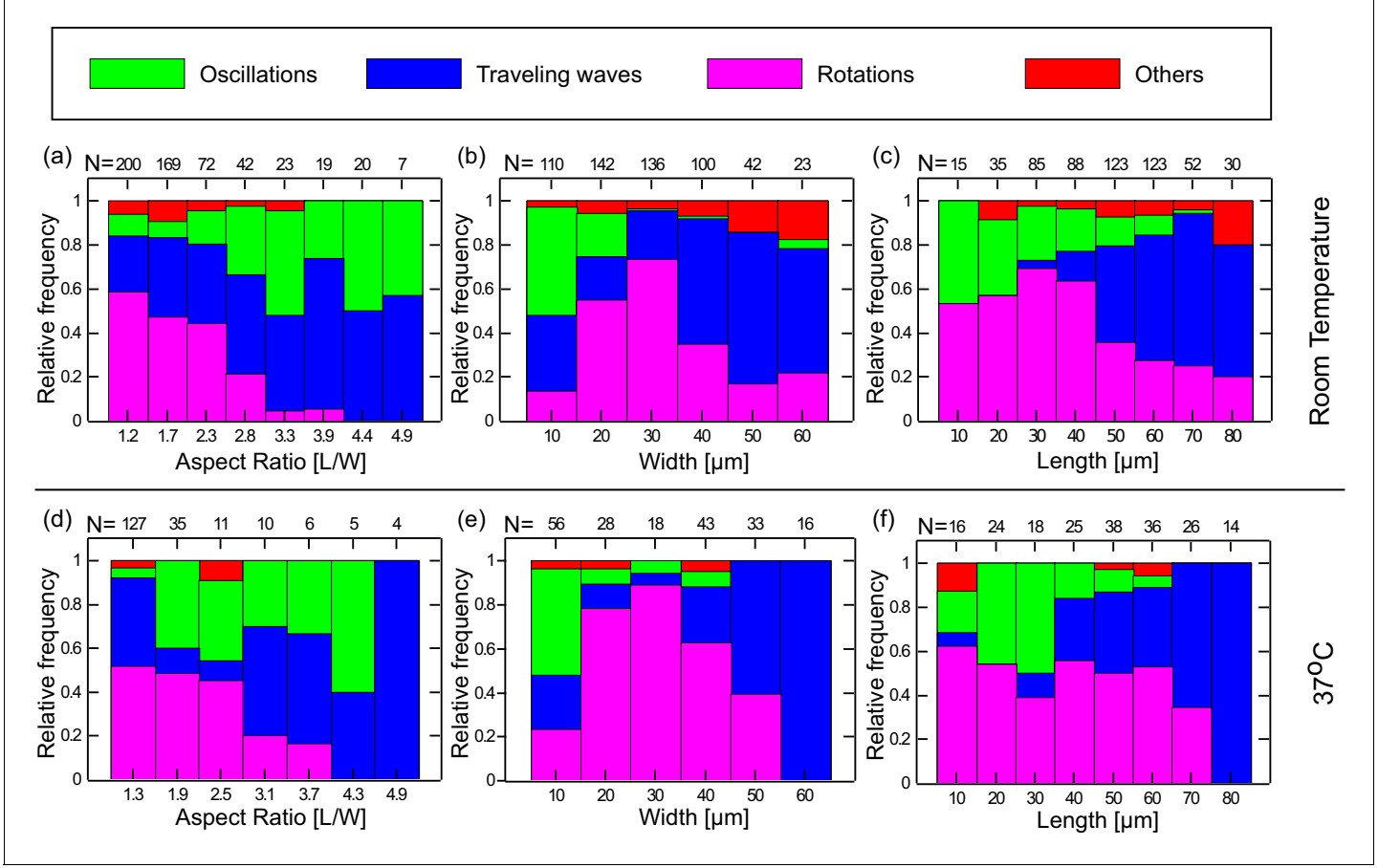

**Figure 3.** Preferred dynamical Min behavior as a function of the chamber-geometry parameters. (a–c) Geometry selection at room temperature. (a) Selection according to the chamber aspect ratio. (b) Selection according to chamber width. (c) Selection according to chamber length. All measurements were preformed at room temperature on a DOPC:DOPG (67:33) SLB supplemented with 0.03 of TopFluor Cardiolipin. The number of chambers observed with the specific geometrical characteristics are indicated above each bar. (d–f) Same as (a–c) for measurements that were preformed at 37°C on an *E. coli* polar lipid extract SLB. The analysis is based on the fluorescence signals of MinE.
The following figure supplement is available for figure 3:

**Figure supplement 1.** Min patterns geometry selection rules based on chambers size.

pattern formation. It also supports the notion that the existence of the connector lines in our setup adds only second-order effects. Thus, by using the soft fabricated valves, we were able to extract and study, for the first time, the geometrical selection rules of the Min system in truly confined 3D structures *in vitro*.

## Phase Diagram of Min pattern in microchambers

Having established a relation between the geometry of the chambers and the spatio-temporal behavior of the Min system, we constructed the phase diagram of the Min patterns, see *Figure 4a*. In this phase diagram, each tile represents, in its color, the most prevalent Min behavior for the chambers with the designated specific dimensions. We restricted ourselves to cases for which we had data from at least 4 different chambers with this specific geometry; the typical number are 20–30 chambers per tile, see *Figure 4—figure supplement 1* for the exact numbers. As can be seen from *Figure 4a*, the phase diagram is nicely separated into three different regions: (i) The majority pattern is rotations; (ii) oscillations are the most prevalent pattern in narrow chambers, in line with previous observation in semi-3D-confined structures (*Zieske and Schwille, 2013*, *2014*); and (iii) as expected (*Loose et al., 2008*, *2011a*), the most prevalent behavior for large and long chambers is

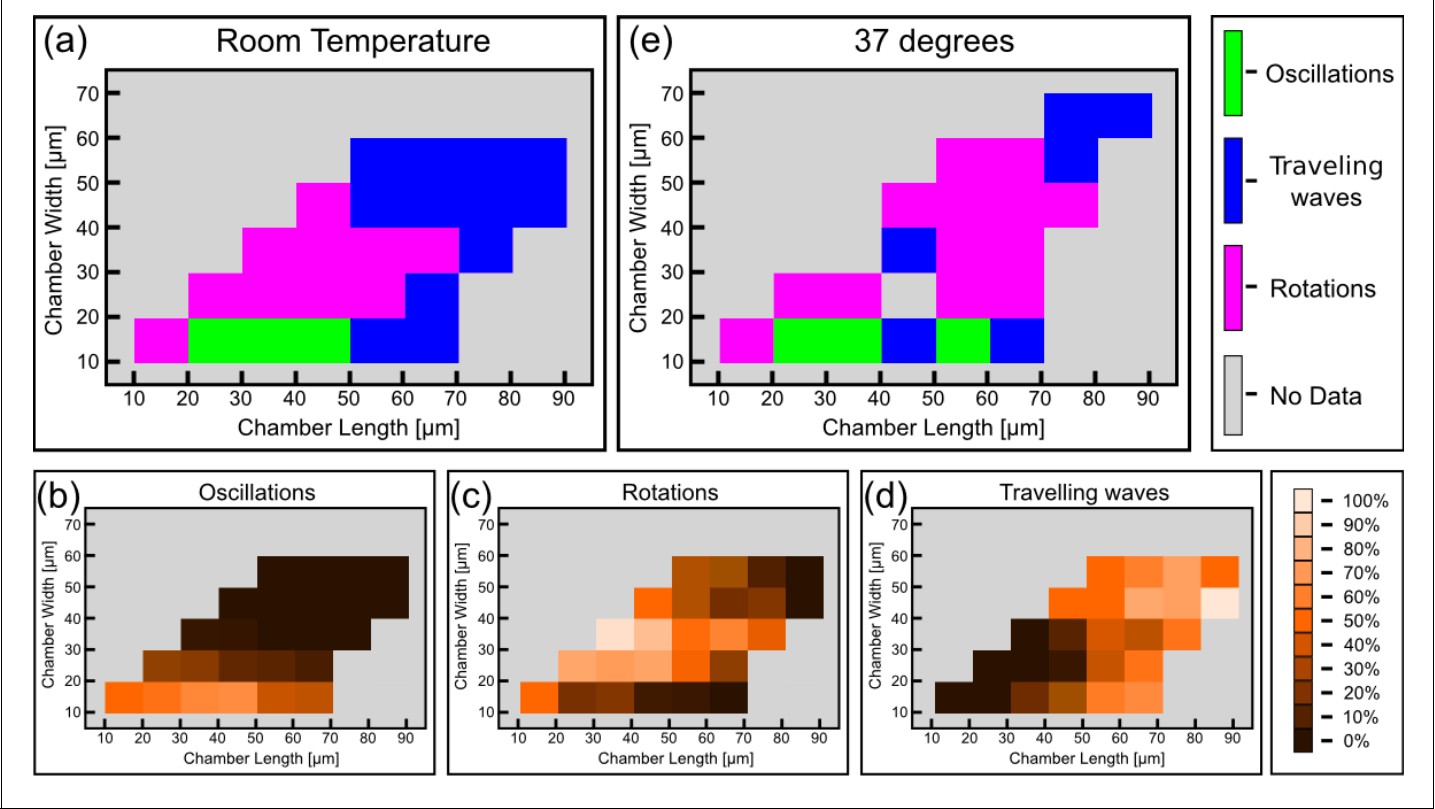

**Figure 4.** Phase diagram of the Min patterns. (a) Phase diagram of the Min behavior at room temperature. Each square represents the most abundant behavior in that geometry. Color code is shown on the right. (b–d) Relative abundance of the pole-to-pole oscillations, rotations and traveling waves in each chamber size. Color code for (b–d) is shown on the right. (e) Phase diagram of Min behavior at 37°C. Color code is the same as in (a). The analysis is based on the fluorescence signals of MinE.

The following figure supplements are available for figure 4:

**Figure supplement 1.** Number of chambers that were used per each tile for constructing *Figure 4a–d*.

**Figure supplement 2.** Detailed analysis of the phase diagram at elevated temperature (Figure corresponding to *Figure 4e*).

that of traveling waves. The relative percentage of the oscillations, rotations and traveling waves is presented respectively in *Figure 4b–d*. The most striking, and unexpected, result of this study is that in fully confined 3D chambers, a large part of the phase diagram is occupied by rotational behavior, i.e., Min patterns that rotate around a fixed point in a spiral fashion. In other words, confining the Min system in 3D chambers mainly results in the formation of spiral waves. When the chambers becomes too narrow, these rotational centers are less stable, presumably due to reflection of the Min concentration front from the chamber boundaries, and oscillations are formed. Oscillations thus appear to be a derivative phenomenon that results from destabilization of spiral rotating patterns by the chamber walls. In the other extreme, for very large areas, the interaction between multiple rotational centers will equilibrate in such a way that the stable behavior becomes a traveling wave.

## Concentration of the Min proteins in the microchambers.

Theoretical models of the Min system rely on the concentration of the MinD and MinE proteins as an important control parameter. Determining the exact concentrations in the microchambers needs some considerations and is less trivial than it may appear to be at first glance. While Min proteins were injected at a well-defined concentration (1 μM MinE and 1.08 μM MinD), the final concentration of the proteins in the chambers are higher than those introduced. The reason is that during the

injection process, MinD molecules will bind the membrane, followed by MinE molecules, while proteins continue to flow into the chamber with the fresh bulk solution. This results in larger final concentration in steady state. We therefore measured the concentration of the final Min proteins inside our chambers using a green fluorescence protein (GFP) calibration. We measured the relative fluorescence of GFP, MinD-Cy3, and MinE-Cy5 in bulk using a fluorometer, yielding a calibration curve of intensity versus concentrations. Next, we injected the regular Min proteins mixture, which contains MinD-Cy3 and MinE-Cy5, together with GFP into the chambers, and measured the resulted fluorescence of the two labeled proteins verses that of GFP on a widefield microscope with a 20X objective. From that, one can infer the actual concentrations (see *Figure 5—figure supplement 1a–c* for the fluorometer calibration curves and the Materials and methods section for the detailed derivation of Min proteins concentration in the chambers). Note that since GFP does not bind the membrane, its concentration in the microchambers is always known. Thus, it can be used as a calibration reference to estimate the concentration of the Min proteins by comparing their relative fluorescence in the microchamber to the relative fluorescence for the case where there is no membrane binding (i.e. in the fluorometer cavity).

The concentration of the Min proteins was measured for 52 different chambers. As can be seen (*Figure 5a*), the actual concentration of the Min proteins in our chambers was significantly (~factor 5 higher than the value for the injected stock solution. Furthermore, a wide distribution is observed, particularly for MinE. Note that we did not observe a relation between the chamber size and the measured concentration of the Min proteins. From these measurements we concluded that the concentration of MinD in our chambers was $4.5 \pm 0.5$ μM (mean $\pm$ SD), the concentration of MinE $6 \pm 3$ μM, and the average ratio of [MinE]/[MinD] amounted to $1.3 \pm 0.5$.

## Wavelength of the Min waves in microchambers

It is of interest to know if transforming the topology from surface 2D (*Loose et al., 2008*; *Ivanov and Mizuuchi, 2010*; *Loose et al., 2011a*; *Vecchiarelli et al., 2014*) or semi 3D (*Zieske and Schwille, 2013*, *2014*) to a fully confined 3D topology will substantially change the wavelength of the traveling waves. This question is interesting since, as mentioned in the introduction, one of the open questions regarding the Min system relates to the differences between its spatial length scale *in vitro* verses *in vivo*. Mean and SD of the measured wavelength of 35 traveling-waves in confined chambers are shown in *Figure 6a*. We measured a wavelength of $43 \pm 6$ μM at room temperature. This value is significantly lower than the wavelength of $78 \pm 12$ μM (n = 30 for traveling waves that we measured on 2D flat SLBs using the exact same protein and lipid composition of the SLB as was used in our microchambers experiments. This shows that the 3D full confinement of the Min system has a clear effect not only on the geometry-selection properties, but also on the characteristic length scale of the system.

Similarly, it is of interest to ask whether the confinement has any influence on the propagating velocity of the Min front. We compared the velocity of all different Min patterns in the chambers (n = 333 chambers) to the propagation velocity of Min waves on the flat surfaces (n = 30, see *Figure 6b*). While on flat SLB surfaces we measured v = $0.6 \pm 0.2$ μm/s, similar to previous measured values (*Loose et al., 2011a*), inside the chambers we obtained a propagation velocity of $0.3 \pm 0.1$ μm/s (while the propagation velocity was $0.4 \pm 0.1$ μm/s for waves only; n = 35). The reduced wavelength inside the chambers was thus accompanied with a reduced velocity.

## Wavelength of the Min system in crowded environments.

We examined the effects of molecular crowding on the characteristics of the waves in our 3D confined chambers. Crowding the liquid environment can have multiple effects. On the one hand, it increases the viscosity of the solution, thus, by Einstein's relation, reducing the diffusivity of the proteins. This parameter is particularly important in reaction-diffusion processes, such as the Min dynamics, since the diffusion rate of the fastest species determines the distance between two maxima of the pattern. On the other hand, the use of a high concentration of molecular crowder (polymers or proteins) can have unintended side effects and influence the stability and binding properties of the studied proteins themselves (*Kuznetsova et al., 2014*). In order to maximize the viscosity of the solution while minimizing the side effects, we employed a solution of 4% PEG 8000, 4% Ficol 400, and 4% BSA (all three commonly used crowders). The measured viscosity of this solution is

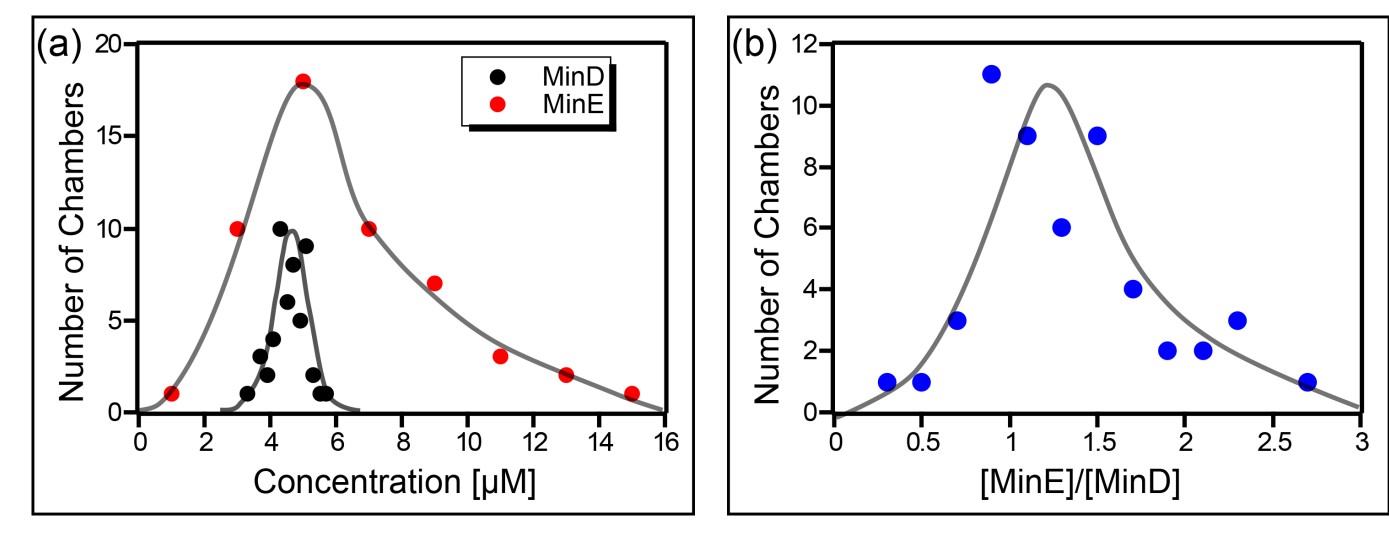

**Figure 5.** Concentration of the Min proteins. (**a**) Histogram of the deduced concentration of MinE and MinD in the chambers. (**b**) Idem for the ratio of MinE to MinD. Lines are guides to the eye. Concentration was measured using a GFP protein as a standard as described in the Materials and methods.
The following figure supplement is available for figure 5:

**Figure supplement 1.** Calibration curves for the fluorescence intensity of (**a**) MinD, (**b**) MinE, and (**c**) GFP, as collected with a fluorometer.

9.9 ± 0.05 cP, i.e., one order of magnitude larger than that of water. Though the exact viscosity inside *E. coli* cells is unknown, this value is close to the estimated one (*Trovato and Tozzini, 2014*). The results (n = 17) for the wavelength and wave velocity of traveling Min waves in confined 3D chambers under these conditions are shown in *Figure 6a and b*, see also *Figure 6—figure supplement 1* for examples of Min waves under viscous conditions. Increasing the viscosity of the environment resulted in a factor 2 reduction of the wavelength from 43 ± 6 μm to 23 ± 4 μm. This reduction in the wavelength is comparable to the reduction in wavelength that was observed by Martos et al. in the presence of 140 g/l Ficol (*Martos et al., 2015*). Next to the reduction in the wavelength, we observed a very large decrease, of more than one order of magnitude, in the wave velocity from 0.3 ± 0.1 μm to 0.02 ± 0.014 μm/s. Note that this substantial decrease in the velocity is much larger than what was observed by Martos et al. and is probably related to the different constellations of 2D surfaces (used by Martos et al.) in comparison to 3D confined microchambers (in this work).

## Dynamics of the Min system at elevated temperature.

The previous results showed that there are various ways to modulate the velocity and characteristics length scale of the Min proteins, and prompted us to look for other means to modulate these properties. We therefore studied the Min system properties in our chambers at an elevated temperature of 37°C. As can be seen in *Figure 3d–f* and *Figure 3—figure supplement 1b*, the geometry selection properties of the Min system at the elevated temperature of 37°C are very similar to those that were observed at room temperature (n = 198): (i) at low aspect ratio of the chambers (L/W), the prevalence to obtain rotational patterns is high while it decreases as the aspect ratio increases; (ii) oscillatory behavior is observed mainly for narrow chambers; (iii) traveling waves become dominant for long and large chambers. The phase diagram at elevated temperature is shown in *Figure 4e* (a detailed analysis of the phase diagram is shown in *Figure 4—figure supplement 2*). The overall picture remains the same, although more scatter is apparent in the data.

The wavelength at the elevated temperature (see *Figure 6a*, n = 34), shows only a small decrease relative to the wavelength at room temperature, that is from 43 ± 6 μm to 37 ± 9 μm (two sample t-test with different variance = 0.001 at β = 0.05). This contrasts the behavior of waves on flat 2D SLBs surfaces, for which we observed a much more profound effect (*Figure 6a*, n = 27). In this case

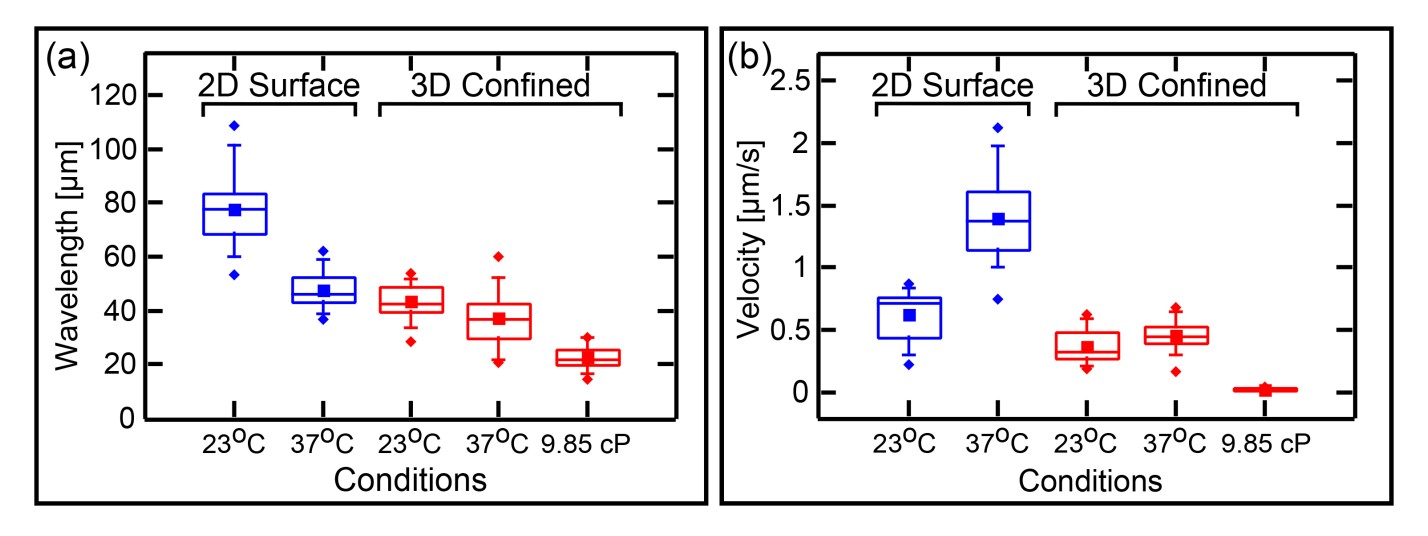

**Figure 6.** Wavelenght and Min patterns front propagation velocity. (**a**) Statistical box plot representing the wavelength of the Min waves at different conditions: at room temperature and at 37°C, at flat SLB (blue) as well as inside the 3D confined chambers (red), and with high viscous medium in the chambers (at room temperature). The lines of each box represent the location of the 25, 50, and 75 percentiles. Full squares represent the location of the mathematical mean. Whiskers represent the 5 – 95 percentile range and the diamonds the minimum and maximum values of the data. (**b**) Velocity of Min pattern propagation at the same conditions. Box representation is the same as in (**a**). The analysis is based on the fluorescence signals of MinE.

The following figure supplement is available for figure 6:

**Figure supplement 1.** Examples of traveling waves in the chambers in the presence of a viscous media at 9.85 cP.

the wavelength reduced from 78 ± 12 at room temperature to 48 ± 6 μm at 37°C. These data thus show that, in confined chambers, confinement is the major cause of the reduced wavelength, with the elevated temperature adding only a small further reduction in the characteristic spatial scale of the system.

Interestingly, we saw an increased, rather than a decreased, velocity of the Min proteins at high temperature. Results are shown in *Figure 6b*, measured from 162 chambers. At T = 37°C, the velocity v = 0.5 ± 0.3 μm/s (velocity for waves only v = 0.5 ± 0.1 μm/s n = 34). This compares to v = 1.4 μm/s (*Figure 6b*) for waves propagating on flat SLBs surfaces. Thus, similar to the conditions in vivo, where the period of the oscillations is inversely correlated with the temperature (*Touhami et al., 2006*), elevated temperature increases the velocity of the Min propagation *in vitro*, doubling from 23°C to 37°C both on 2D surfaces and in 3D confinement. However, in contrast to the in vivo behavior, the *in vitro* wavelength is also temperature dependent. Since the viscosity of the bulk media change only by a factor of 1.3 from 23°C to 37°C, we have to attribute these results to a change in one of the reaction parameters of the reaction-diffusion Min system. We can conclude that there are different ways to modulate the different dynamical characteristics (wavelength and velocity) of the Min system.

## Discussion

In this paper we report a comprehensive experimental data set of Min patterns in fully confined fluidic chambers that are internally coated on all surfaces with a supported lipid bilayer. The ability to obtain a detailed picture of the Min patterns *in vitro* is important for three reasons. First, the theoretical understanding of Min pattern *in vivo* is still incomplete, and *in vitro* studies with their exquisite control will help to resolve existing questions such as the origin of the symmetry-breaking mechanism. Second, the development of a comprehensive theoretical framework for the Min system behavior depends on the ability to experimentally compare the *in vivo* and the *in vitro* cases under well-defined conditions such as dimensionality and size. Third, applications that aim at utilizing the

Min system for engineering complex behavior in synthetic cells and other man-made systems depend on the ability to fully control its behavior *in vitro*.

Previously, Min proteins were reconstituted *in vitro* (*Loose et al., 2008*; *Ivanov and Mizuuchi, 2010*; *Loose et al., 2011a*; *Schweizer et al., 2012*; *Martos et al., 2013*; *Vecchiarelli et al., 2014*; *Zieske et al., 2014*; *Martos et al., 2015*; *Vecchiarelli et al., 2016*), and Min oscillations were observed in fabricated microchambers (*Zieske and Schwille, 2013*, *2014*). In the latter case, the microchambers had a half-open configuration (i.e., with a top surface not coated with SLB) with a limited width (~10 μm), a height of 10 μm, and varying lengths leading to a range of aspect ratios (1.2–24). In our case, we studied the system for microchambers with a height of 2.4 μm, a width of 10–60 μm, and a length of 10–80 μm. We thus considerably expand previous experimental data since we mapped the Min behavior in a well-controlled manner over a section of the geometrical conditions where the microchambers have a much broader range of widths, while tightly restricting the height of the microchambers. The novelty of our approach can particularly be appreciated from the fact that, while previously no ordered patterns were observed in chambers wider that 10−20 μm (*Zieske and Schwille, 2013*, *2014*), we observed a well-defined behavior of the system that was mapped into an ordered geometric phase diagram, as discussed below.

In this article, we have shown that total confinement of the Min system within 3D chambers leads to three main type of patterns: (i) rotational patterns in the form of spiral waves, which are the majority of the patterns found; (ii) periodic oscillations that occur mainly if the chamber width is small in comparison to the typical spatial length scale of the system *in vitro*; (iii) traveling waves that occur mainly if one of the dimensions of the chamber is larger than the typical spatial length scale of the system.

In our microchambers, rotational behavior in the form of one or multiple adjacent spiral waves thus contributes the largest fraction of the phase diagram. Spiral waves are common in various biological and chemical reaction-diffusion processes such as *Dictyostelium discoideum* aggregation, calcium variations in *Xenopus laevis* oocytes, and the famous Belousov-Zhabotinsky (BZ) reaction (see *Epstein and Pojman, 1998* and references therein). They were also observed for the Min system on flat SLBs (*Loose et al., 2008*; *Vecchiarelli et al., 2014*). For reaction-diffusion systems other than the Min system, various symmetry-breaking-mechanisms were invoked in order to explain this spiral behavior. For example, for the BZ reaction, spiral waves were attributed to a Hopf bifurcation mechanism (*Hagan, 1982*). In addition, it is well known that in nonlinear reaction-diffusion systems, several symmetry-breaking-mechanisms may coexist in different parts of the parameter phase space and, as result, a plethora of static or dynamic patterns can emerge (*Yang et al., 2002*). Many times these patterns possess similar observable behavior, yet having different underlying symmetry-breaking-mechanisms. In light of this knowledge, it is well possible that *in vitro* Min patterns, in fact, emanate from a somehow different symmetry-breaking-mechanism than the Min oscillations in live cells despite the fact that the proteins are the same and the patterns share various similar characteristics with the *in vivo* case. In other words, in spite of the common belief that the main difference between the *in vivo* behavior of the Min system and its behavior *in vitro* is merely related to a different wavelength, possibly, the symmetry-breaking-mechanisms are not exactly the same in both cases. The existence of spirals *in vitro* and their abundance in our confined microchambers, when compared to their absence from *in vivo* observations where pole-to-pole oscillations dominate (*Wu et al., 2015*), may be related to this.

Note that a complete understanding of a biochemical system depends on two complementary sources: a theoretical model that can predict the system behavior and detailed description of its behavior under as wide as possible experimental conditions. For a complex non-linear system, like the Min, it is relatively easy to construct a model that predicts the system behavior under a limited subset of the experimental data. This task becomes much harder, however, when experimental data exist for a wide range of conditions and system geometries. Such data thus help to restrict the class of possible theoretical models. In this report we provide such a detailed description of the *in vitro* behavior of the Min system in a wide section of the geometrical parameter space. Thus, our data restrict the possible classes of theoretical models that can explain the *in vitro* behavior of the Min system. In addition, since our microchambers height is relatively close to the actual diameter of a bacterial cell, our data further help to restrict the possible models for symmetry-breaking-mechanisms that can describe the system behavior *in vitro* as well as *in vivo*.

We observed that the spiral patterns were found not to be stable in narrow chambers that are smaller than, or equal to, the characteristic 10 µm length scale of spirals. We observed that pole-to-pole oscillations are established as the chamber walls restrict spirals to form. In other words, *in vitro* oscillations in fact appear to be a form of truncated spirals. This demonstrates how the interplay between geometrical confinement and the intrinsic spiral symmetry-breaking-mechanism of the Min system may produce distinct patterns. These data are the first to suggest that *in vitro* oscillations are truncated spirals, a fact that further shows the importance of studying the geometric phase diagram of the Min system, as was done here. Please note also that *in vivo*, spiral rotations were not observed (*Wu et al., 2015*). Neither were they suggested as the origin of the symmetry-breaking mechanism in theoretical works. This serves as an additional note of caution for drawing a too simple correspondence between oscillatory patterns that are observed *in vitro* and those observed in vivo.

Our findings are further corroborated by comparing the nature of oscillations *in vitro* to those that are formed *in vivo*. To study this difference, we collected movies of *in vivo* oscillations (see *Supplementary file 1* and *Supplementary file 2*). By comparing the kymographs in both cases (*Figure 2e*, *Figure 2—figure supplement 1* and *Supplementary file 1e–g*), one can clearly see the distinctly different behavior in these two cases. *In vivo*, the pole-to-pole oscillations amount to the establishment of polar zone that is replaced by a second polar zone near the opposite pole within a short time scale. In contrast to this, *in vitro* oscillations are more akin to a traveling wave that propagates toward one pole, whereupon it is then replaced by a wave that is established close to the chamber middle and propagates toward the opposite pole (we acknowledge the anonymous referee three for pointing out this difference). Similar behavior was also observed in *in vitro* oscillations of other reconstitution work (*Zieske and Schwille, 2014*). This may also be the reason why steep *in vitro* Min gradients that would restrict FtsZ localization to a single ring could not have been established. Note that *in vitro* we report the MinE signal while in vivo we imaged MinD (as the MinE signal was low in this strain). Essentially this does not alter the conclusion that is drawn here since in this strain the MinE signal merely follows the MinD signal (see [*Wu et al., 2016*]).

Why do traveling waves form the majority pattern in very large chambers while spirals are less abundant? *In vivo*, Min waves are hardly detected. Shih et al. reported the formation of transient Min waves in *E. coli* cells harboring the MinE$^{D45A/V49A}$ mutation that prevents the dimerization of MinE (*Shih et al., 2002*). Recently, Bonny et al. also reported traveling waves in filamentous cells, but the abundance of this phenomenon was not reported (*Bonny et al., 2013*). In this last case, the Min operon was induced at a saturating concentration and thus the concentration of MinDE was probably much higher than in the native case for *E. coli* (*Sliusarenko et al., 2011*). Finally, in vivo traveling waves were also observed in a small minority of filamentous cells, even if the Min operon is not over-induced and when the Min proteins do not harbor any mutations (see *Supplementary file 1d* and *Supplementary file 2*). Note that in all these cases, the *in vivo* traveling waves represent a form of abnormal behavior which occurs if the MinE function is impaired or in other anomalous cases. This contrasts to the *in vitro* case, where traveling waves can be a generic feature of the Min system. In contrast to sporadic reports of Min waves *in vivo*, waves are almost always detected in *in vitro* assays on 2D SLBs. Two reasons are responsible for this observation. First, on flat 2D SLBs the spirals waves are not confined and hence waves that emanate from one center can travel long distance and interfere with waves that emanate from a second spiral center to form traveling waves. In our chambers, when their area became large, the intrinsic Min pattern formation mechanism and the reflection from the chambers walls resulted in the annihilation of spiral centers and the establishment of traveling waves, similar to the 2D case where boundaries are absent. The second reason for the formation of traveling waves is probably related to the concentration of the proteins in the *in vitro* assay. In a typical *in vitro* assay, one usually controls the bulk concentration of the proteins as the supply material. However, due to the small surface-to-volume ratio in typical *in vitro* assays, the actual concentration of the proteins on the membrane and the replenished reservoir of new Min proteins is higher than *in vivo* (cf. the above section on Min concentration in the microchambers). In fact, due to the experimental protocols employed, this observation is also correct for previous reconstitutions of the Min system in grooves (*Zieske and Schwille, 2014*), as well as for published *in vitro* Min system flow assays (*Vecchiarelli et al., 2014*; *Ivanov and Mizuuchi, 2010*). This situation is qualitatively different from the *in vivo* case where, during the formation of polar zone or a Min band, most of the MinD proteins are recruited to the membrane and the cytosolic concentration thus drops to a low value. Indeed, in a very recent publication, Vecchiarelli et al. suggested that *in vitro*

spirals waves are related to high concentration of the Min proteins, while reconstituting the Min system under limiting concentration conditions resulted in a dynamical series of Min bursts that are reminiscent of the behavior in cells (*Vecchiarelli et al., 2016*). Vecchiarelli et al. further checked the behavior of the Min proteins when MinE was replaced by a membrane-binding deficient mutant MinE[11–88], and observed that in this case, Min waves were supported at high protein concentration, but bursting dynamics were absent. This observation was consistent with the one of Zieske et al. (*Zieske and Schwille, 2014*) where oscillations were not observed with a similar mutant MinE(Δ3–8). Thus, it seems that the interrelations between several factors determine the pattern formation mechanism *in vitro*: the surface-to-volume ratio, the protein concentration, the confinement, and the binding of MinE to the membrane. Our results, with the relatively high Min protein concentration that we measured, support their conclusions. The fact that we fully confined the Min system in 3D chambers with a relatively small height enabled us to look on the behavior of the system in conditions with a surface-to-volume ratio that is much larger than previously studied. Combined with our measurements of the total Min protein concentration in the chambers, our results set a lower limit for the protein concentration that still reproduces traveling waves, rotational spirals, or oscillations under these geometric conditions. It will be interesting to check if current models of the Min system that reproduce in vivo behavior will also be able to reproduce traveling waves by simply increasing the proteins concentration. Indeed, Bonny et al. were able to show traveling waves in their theoretical system by increasing the Min concentration by 63% or by increasing the MinE to MinD ratio (*Bonny et al., 2013*). In our geometrical analysis of Min patterns selection, we showed that in chambers with high surface-to-volume ratio, and a high Min proteins concentration, the cumulative effect of annihilation of multiple spirals and of the reflection from the walls can result in the formation of traveling waves.

A related question regards the Min wavelength. We have observed a reduction in the Min wavelength by a factor of two in our microchambers relative to the case of 2D flat SLB. Taking a reaction-diffusion point-of-view, this observation can be explained in two ways. First, the lateral confinement by itself may be responsible for choosing dynamical modes of the system with a smaller wavelength. Second, the height reduction could change the balance between the reservoir of the free Min proteins in solution relative to the bound ones. Indeed, based on their model, Halatek and Frey recently suggested, that reducing the volume of Min proteins above the surface will drive the typical wavelength of the system to lower values (personal communication). Similarly, it was shown theoretically by Thalmeier et al. (*Thalmeier et al., 2016*) that even in the absence of MinE, *Arabidopsis thaliana* MinD can form an intracellular gradient just based on the relations between the bulk diffusion and nucleotide exchange rates, but only in highly confined spaces. Thus, our observation that reducing the height of the chamber is accompanied by a reduction of both the wavelength and the wave velocity further stresses the value of our approach of studying the Min behavior in 3D confined spaces with a high surface-to-volume ratio. We note, however, that our chambers are 2.4 μm high, not too much larger than the 1 μm diameter of an *E. coli* bacterium, and yet, we measured a wavelength that is an order of magnitude larger than the characteristic length scale observed in vivo.

We have shown that two factors can lower the wavelength further. First, the Min wavelength decreased by a factor of 2 when the viscosity of the medium was raised to 10 cP, i.e., similar to that of cells and ten times larger than a regular buffer. Still, even at this high viscosity, the wavelength was larger by a factor of ~5 from the one measured in vivo. In addition, the reduction in wavelength was accompanied by a very large reduction in the wave propagation velocity. Note that the measured velocity in the highly viscous media was similar or even smaller than the one measured *in vivo* (*Unai et al., 2009*) (see also the supporting information in [*Bonny et al., 2013*]). It is thus unlikely that additional increases in viscosity can bring both the characteristic length scale and the propagation velocity of the system *in vitro* to the *in vivo* values. Interestingly, unlike the situation in unconfined 2D reconstitution of the Min proteins (*Martos et al., 2015*), in microchambers with a high surface-to-volume ratio, the period of the Min wave (i.e., wavelength/velocity) was not constant but was reduced by a factor of about 8.

Second, we showed that increasing the temperature also reduced the wavelength. The effect of the temperature on the characteristic wavelength was much larger for the Min waves on flat SLBs than in the 3D confined chambers. Since the diffusion rate does not change substantially between room temperature and 37°C (only a factor of 1.3, by the Einstein's relation), the temperature effect in the 2D surface case is most likely the result of a change in one of the reaction parameters, most

probably the ATP hydrolysis rate. This situation is different in the microchambers. Our observations suggest that the main *in vitro* determinant of the wavelength at elevated temperature in confined spaces is the geometrical confinement itself. In other word, geometrical confinement forces the system to choose a reaction-diffusion mode with a reduced wavelength which can only marginally be tuned further by increasing the temperature. Thus, in highly confined spaces the effect of temperature on the characteristic scale of the Min system is largely diminished, and is accounted by the small increase in diffusion rate. This fact can explain the previous *in vivo* results where no temperature dependence for the wavelength was detected (*Touhami et al., 2006*). It should be noted that the reduced length scale was accompanied by an increase, rather than a decrease, of the wave velocity. This observation is in line with the *in vivo* case where the oscillation period was smaller at high temperature. Our results points out that the dynamical characteristics of the Min system contain several decoupled parameters, such as wavelength and velocity that can be tuned in opposite directions. It is thus essential to study their action in the context of 3D confined spaces with a high surface-to-volume ratio.

The symmetry-breaking mechanism of the Min pattern formation depends on a combination of all its dynamical parameters. We have shown that dynamical aspects of the system can be tuned in at least three different ways: by confinement, by changing the temperature, and by reducing the bulk viscosity. Yet, since in our 3D confined structures, even at elevated temperature or with high viscous media, the *in vitro* Min behavior did not quantitatively reproduce the *in vivo* one, probably a change of yet another dynamical parameter is necessary. This can be a simple thing like a change in the concentration of one of the Min proteins (*Vecchiarelli et al., 2016*) or a change in one of the reaction rates. It is also possible that yet another physical or biological mechanism contributes to the different symmetry breaking in both cases. Biochemical assays have shown intricate effects of the Min system on the membrane organization in the *in vivo* context. For example, MinD increased the order of the lipids and decreased their mobility in inverted inner *E. coli* membranes that contained integral proteins more than it does in synthetic vesicles that are purely lipidic (*Mazor et al., 2008*). Similarly, the Min system affects the association of inner-membrane peripheral proteins and interacts with some of them directly (*Lee et al., 2016*). It is thus possible that a yet unidentified protein species is needed in order to reproduce the in vivo geometrical selection rules of the Min system in an *in vitro* environment.

To sum up, we have studied the geometry selection rules of the Min system in 3D fully confined chambers. We found three main patterns in these confined chambers: spiral rotations, oscillations and traveling waves. Spiral behaviors were the most abundant ones in a large part of the phase diagram and we suggest that both traveling waves and oscillations result from interrelation between the spiral symmetry-breaking mechanism and the effects of confinement.

## Materials and methods

### Materials

1,2-dioleoyl-sn-glycero-3-phosphocholine (DOPC), 1,2-dielaidoyl-sn-glycero-3-phospho-(1'-rac-glycerol) (DOPG) and *E. coli* polar lipid extract were purchased from Avanti polar lipids. Tris was purchased from Promega. Potassium chloride and imidazole were purchased from Merck. Silicon wafers were from universitywafers.com. RTV 615 PDMS was purchased from Momentive. Nitric acid was from Merck. Cy3-NHS and Cy5-maleimide dyes were purchased from GE Healthcare. Phosphoenol-pyruvic acid (PEP) was from Alfa Aesar. All other materials were from Sigma-Aldrich unless otherwise stated.

### Fabrication of structures on silicon wafers

To fabricate the lower layer of the chip, which consists of structures with three different heights, a silicon wafer was processed in three different steps. First, a 4″ wafer was cleaned with a nitric acid for 10 min under sonication, washed with water and then dried. PMMA 495K 8A (MicroChem) was spin-coated on the wafer at 500 rpm for 5 s and then at 3000 rpm for additional 55 s. Next, the wafer was baked for 1 hr at 180°C and the chambers pattern that was designed using Klayout (RRID:SCR_014644) was written on the PMMA layer using a Vistec EBPG 5000+ (acceleration voltage 100 kV, aperture 400 µm, dose of 800 µC/cm$^2$ and a resolution of 100 nm). Altogether 9 similar chamber

structures were written on the wafer. The written PMMA layer was developed in MIBK: isopropanol (1:3) for 9 min, was washed for 30 s in isopropanol (IPA) and the wafer was spin dried. A Bosch deep reactive-ion etching process, with an inductive coupled plasma (ICP) reactive-ion etcher (Adixen AMS 100 I-speeder), was used to etch the structures into the silicon wafer. The process consisted of alternate etching (sulfur hexafluoride, $SF_6$) and passivation (octafluorocyclobutane, $C_4F_8$) cycles. During the process the pressure was kept around 0.04 mbar, the temperature of the wafer was kept at 10°C, while the plasma temperature was 200°C. The sample holder was held at 200 mm from the plasma source. The etching step involved 200 sccm $SF_6$ for 7 s with the ICP power set to 2000 W without a bias on the wafer itself. The passivation step was done with 80 sccm $C_4F_8$ for 3 s with the ICP power set to 2000 W and the bias power on the wafer alternate with a low frequency: 80 W, for 10 ms, and 0 W for 90 ms. Total etching time was 34 s. After etching, the wafer was cleaned with Nitric acid for 10 min with sonication.

Next, the connector lines were fabricated on the same wafer via similar steps with the following small modifications. Spincoating of PMMA was done at 3000 rmp in the second step. Baking was done for only 45 min. The dose was 742 $\mu C/cm^2$ and the resolution was 15 nm. Development was done for 3 min. After development the PMMA was descummed on Tepla machine at 0.6 mbar at 300 W with 100 sccm for 1.5 min. Dry etching was done on the same Adixen AMS 100 I-speeder machine. The first step was the same as previously described only that the $SF_6$ step lasted 2.4 s and the passivation step lasted 1 s. The total etching time was 10 s. Next, another dry etching step was applied, this time the ICP power was set to 250 W, the bias power was 20 W, the source-target distance was 240 mm, pressure was kept around 0.04 mbar and the temperature was kept at 10°C. The gas combinations was $SF_6$ 200 sccm, Ar 100 sccm and $O_2$ 100 sccm. Total etching time was 5 min. The wafer was then cleaned in the Adixen AMS 100 I-speeder machine at a pressure of 0.04 mbar, ICP power of 2500 W with a biased power of 60 W, a source-target distance of 200 mm and a temperature of 10°C, using a $O_2$ gas at 200 sccm for 5 min. The wafer was finally cleaned in Acetone at 45°C for 10 min and in fuming nitric acid for 10 min with sonication.

The reservoirs were fabricated similar to the chambers with the following modifications: the spincoating was done at 1000 rpm, the e-beam writing resolution was 150 nm with a dose of 889 $\mu C/cm^2$. Development was done for 12 min. Etching was done for 320 s. Final cleaning was done at 45°C for 10 min in acetone following by cleaning step in nitric acid for 10 min with sonication.

To fabricate the upper valves, a 4″ wafer was cleaned with a nitric acid for 10 min under sonication, washed with water and then dried. Next, a thin layer of hexamethyldisilazane (BASF SE) was spincoated (1000 rpm for 1 min) and baked at 200°C for 2 min. The negative e-beam resist NEB22A (Sumitomo Chemical Co., Ltd) was spin-coated (1000 rpm for 1 min) and the wafer was baked at 110°C for 3 min. The structures were written similar to the chambers that were described above with a resolution of 100 nm and a dose of 20 $\mu C/cm^2$. The structures were immediately developed in MF322 (Dow Chemical Company) for 45 s following by a moderate development with MF322:water (1:9) for 15 s and a washing step in water for additional 15 s. Etching was done using the same reactive ion etching process that is described above for 310 s. Finally, the wafer was cleaned with nitric acid for 10 min under sonication.

After fabrication of both wafers (the one with the lower layer and the one with the upper layer), they both were rendered hydrophobic by placing the wafers for at least 12 hr in a desiccator at a pressure of ~0.6 mbar together with 30 $\mu$l of (tridecafluoro-1,1,2,2-tetrahydrooctyl) trichlorosilane (abcr GmbH and Co.). This treatment forms a hydrophobic monolayer on the wafers surface.

To study Min patterns on 2D surfaces we fabricated PDMS flow cells with lateral dimension of 3.325 × 2 $mm^2$. First, a silicon wafer was cleaned for 10 min in nitric acid under sonication. Next, a thin layer of hexamethyldisilazane was spincoated (1000 rpm for 1 min) and baked at 200°C for 2 min, following by spincoating of AZ 5214 resist (Microchemicals GmbH) at 1000 rpm and baking step at 105°C for 4 min. The wafer was then exposed on EVG 620 mask aligner (EVG) through a polyester film photomask (JD photo) and was developed with MF321 (Dow Chemical Company) for 4 min following by a washing step in water for 30 s. Next, the wafer was baked for 10 min at 180°C and was etched similar to the chambers pattern with the slight modification that the etching time was 60 min. The depth of the flow channels was 137 °$\mu$m. Finally, the wafer was cleaned using nitric acid for 10 min under sonication and was rendered hydrophobic as described above.

## Formation of the PDMS chips

Since PMMA is a positive resist, the surface of the lower wafer contained grooves that are identical to the ones we like to have on the PDMS chip. In order to get a PDMS layer where the reservoirs, chambers and side channels are grooved inwards we have used the method of double replication of the structures with PDMS. First, 30 g of RTV615 at a volume ratio of 5:1 base:crosslinker was poured on the lower layer wafer and was baked for 4 hr at 60°C following by a baking step for another 4 hr at 120°C. Next, 9 different chips (20 × 20 mm) were cut from the PDMS mold, and served as masters for the second step replication. These masters were treated with (tridecafluoro-1,1,2,2-tetrahydrooctyl) trichlorosilane in the same way as is described above. To form a thin PDMS lower layer containing all the desired structures, RTV615 at a mass ratio of 5:1 base:crosslinker was spincoated separately on each master (500 rpm for 30 s followed by a step at 1400 rpm for 60 s). The PDMS masters with the spincoated PDMS layer were then baked at 60°C for 45–60 min.

Glass coverslips (22 × 22 mm, VWR, thickness no. 1.5) were cleaned by sonication in acetone for 30 min following by a sonication step in isopropanol (IPA) for 30 min and a final wash in MiliQ water. The coverslips and the top layer of the spincoated PDMS were plasma activated for 12 s in a plasma machine (Plasma PREEN I, plasmatic system Inc.) with a flow of 1 SCFH $O_2$. A coverslip was bound on top of each spincoated PDMS layer and was baked for 10 min at 60°C. The coverslip-bound chips were immersed in methanol overnight. Finally, the lower PDMS layers bound to the corresponding coverslips were peeled off the PDMS masters, dried and kept separately in a plastic box.

To form the PDMS layer that separates the upper and lower layers, 5–10 g of RTV615 at a volume ratio of 15:1 base:crosslinkerwas poured on a flat 4″ wafer that was treated with (tridecafluoro-1,1,2,2-tetrahydrooctyl) trichlorosilane as was described above, and the PDMS was spincoated (500 rpm for 30 s followed by a step at 2000 rpm for 120 s) to form a thin layer that was baked at 60°C for 1 hr.

To form the upper PDMS layer, 30 g of RTV615 at a mass ratio of 5:1 base:crosslinker were poured on the upper layer wafer baked at 60°C for 45–60 min, and 20 × 20 mm pieces were cut from the PDMS mass.

Next, the upper PDMS layer pieces and the PDMS separation layer were plasma activated for 12 s at Plasma PREEN I machine with 1 SCFH $O_2$, bound one to the other and were baked at 60°C for 10 min.

Before each experiment, the upper PDMS layer bound to the separation PDMS layer was peeled off the flat silicon wafer, holes were punched and both the peeled piece and a lower layer coverslip-bound chip were plasma activated for 12 s with a Plasma PREEN I machine supplemented with a flow of 1 SCFH $O_2$, and aligned manually one on top of the other using a home-build alignment machine build on a IX71 Olympus microscope that was equipped with a 4X objective (UPIanFLN, N. A. 0.13). Finally, the two parts were bound to each other and were baked at 80°C for 10 min.

For preparing the PDMS chips of the large flow cells that were used for studying the Min dynamics on 2D surfaces, RTV615 at a mass ratio of 10:1 base:crosslinker, was mixed, poured on the silicon wafer master and baked for 1–2 hr at 80°C. Subsequently, the PDMS was peeled from the silicon wafer, a PDMS chip (20 × 20 mm) was cut, holes were punched and both the chip and a coverslip (that was previously cleaned in acetone and isopropanol as described before) were activated for 12 s with a Plasma PREEN I machine supplemented with a flow of 1 SCFH $O_2$. Finally, the PDMS chip was bound to the coverslip and was baked for 10 min at 80°C.

## Purification of Min proteins

Purification of MinD and MinE was done as described before (*Loose et al., 2008*) with slight modification. Briefly, *E. coli* BL21(DE3) cells containing pET28a with either His-MinD or His-MinE were grown in the presence of 50 μM kanamycin in LB media to an O.D. of ~0.6–0.8 at 37°C and 180 rpm shaking. Next, the expression of the Min proteins was induced with 1 mM of IPTG and the cells were grown overnight at 18°C with 180 rpm shaking. The cell were then harvested by centrifugation at 4500 g for 30 min, washed with buffer A (50 mM sodium phosphate pH 8.0 at 4°C, 300 mM NaCl), and then were resuspended in a lysis Buffer (buffer A supplemented with 10 mM imidazole, 5 mM TCEP (tris(2-carboxyethyl)phosphine), a complete protease inhibitor (Roche) (and 100 mM ADP for the MinD case only). Cells were lysed in a cell disrupter machine at 15,000 PSI and the lysate was cleared by centrifugation at 37,500 g for 1 hr. The supernatant was loaded on a 5 ml HisTrap column

(GE Healthcare) on an AKTA machine (GE Healthcare), and the column was washed once with lysis buffer supplemented with 10% glycerol. Next, the column was washed with the same buffer +20 mM imidazole +10% glycerol and with the same buffer +50 mM imidazole +10% glycerol. MinE was eluted with 250 mM imidazole and MinD with 160 mM imidazole. Min proteins fractions were collected, concentrated with amicon ultra 10 kDa (Merck Millipore) and further purified on a Sephacryl $S - 300$ HR 16/60 column on an AKTA machine (GE Healthcare) using a storage buffer (50 mM Hepes pH 7.25 at 4°C, 150 mM KCl, 10% V/V glycerol, 0.1 mM EDTA pH 7.4 and 80 µM of ADP for the MinD case).

Protein concentration was measured using a QuantiPro$^{TM}$ BCA assay kit (Sigma-Aldrich). The ATPase activity of MinD was measured by detecting the reduction in the NADH absorption line at 340 nm. For the activity assay, 100 µl of solution, containing MinD (1–5 µM), was incubated together with MinE (1–5 µM), *E. coli* polar-lipid SUVs (1 mg/ml), Pyruvate kinase (PK) (0.02 mg/ml), ATP (5 µM), Phospho(enol)pyruvic acid (PEP, 5 µM) at 37°C. Negative control assays without MinE, without the liposomes, or without the MinD, were similarly prepared and handled. At specific times (every 40 to 60 min), 4 µl fractions of the activity assay or the control reactions were removed and added to 36 µl containing PEP (2.1 mM), NADH (0.22 µM) and a solution of Lactate dehydrogenase/ PK (Sigma-Aldrich, 22 U of each component). Next, the mixed solutions were incubated at 37 degrees for 10 min and then moved to ice. Finely, the absorption at 340 nm was measured using a nanodrop machine (Data not shown). We used NHS-Cy3 to label MinD and Maleimide-Cy5 to label MinE according to the manufacturer procedure (GE Healthcare). The degree of labeling was $\frac{Cy3}{MinD-Lysine} = 0.88$, $\frac{Cy5}{MinE-Cysteine} = 0.45$.

## Preparation of small unilamellar vesicles

Small unilamellar vesicles (SUVs) were prepared through the common method of thin film hydration. Briefly, lipids in the selected molar ratio dissolved in chloroform were mixed in a round shaped flask (either *E. coli* polar lipid extract or 67:33 DOPC:DOPG supplemented with 0.03 of TopFluor Cardiolipin, except for measuring proteins concentration where the TopFluor Cardiolipin was not added). The chloroform was evaporated using a nitrogen stream and further by incubation in a desiccator for at least 2 hr at a pressure of ~1 mbar. Next SUV buffer (10 mM Tris pH 7.45 at 21°C, 150 mM KCl) was added to a final concentration of 5 mg/ml and the flask was shaken at 250 rpm until all the lipid film completely hydrated. Next, the solution was sonicated at 36°C for ~30 min and was extruded through a 30 nm filter 21 times. Finally, the SUVs were frozen in liquid nitrogen and stored at −80°C.

## Observation of Min patterns

Min protein were observed on a commercial Olympus IX81 microscope equipped with a 60X objective (PlanApoN TIRFM UIS 2, NA 1.45, oil immersion) or with a 20X objective (UPlansApo, NA 0.85, oil immersion), a USHIO USH-1030L mercury lamp, a Mad City lab Micro-drive stage, an Uniblitz VMM-T1 shutter drive and a Hamamatsu C4742-95 12 ERG camera. Microscope was controlled via the Micro-Manager program (*Edelstein et al., 2010*), and the stage was controlled via a self-written program in Labview. For high-temperature experiments, we used a Julabo F12 water-circulating bath that was connected to a custom-designed heating chamber and an objective heater (Live cell instruments, chamlide.com).

In a typical chamber experiment, we first flushed SUVs into the device at a concentration of 2.5 mg/ml in SUV buffer. The vesicles were incubated inside the device at 37°C for 1 hr. Next, the chip was washed with a Harvard apparatus 11 plus pump using a Min buffer (25 mM Tris pH 7.45 at 21°C, 150 mM KCl, 5 mM MgCl$_2$) for about 1 hr at a flow rate of 75–150 µl/h. For room-temperature experiments in the chambers, we used a lipid composition of DOPC:DOPG (67:33) supplanted with 0.03% TopFluor Cardiolipin. For the elevated temperature experiments in the chambers, we used *E. coli* total lipids extract supplemented with 0.03% TopFluor Cardiolipin.

After the chambers were extensively washed, 10–50 µl of Min buffer containing (unless otherwise mentioned): 1.1 µM MinD (either alone or in most of the cases in total divided between 0.9 µM MinD plus 0.2 µM MinD-Cy3), 0.8 µM MinE, 0.2 µM MinE-Cy5, 5 mM ATP (magnesium salt), 5 mM PEP, and 10 µg/ml PK, was injected into the device and the inlet and outlet were blocked using a sellotape. Next, the device was placed under the microscope. After assuring that patterns started to form in the chambers (~10 min after injection), the valves were closed by applying a pressure of 2–3

bars from an argon gas cylinder and the device was incubated for additional of ~1 hr. Finely, the device was scanned and movies lasting ~10 min were recorded. Typically, the frame rate was 0.1 Hz.

For studying Min patterns on 2D supported lipid bilayer we used a similar protocol for preparation and observing the Min patterns to the one that was used for the chamber device with a slight modification that SUVs washing was done manually. In all these experiments we have used SUVs with lipid composition of DOPC:DOPG (67:33) supplemented with 0.03% TopFluor Cardiolipin.

## Analysis of the Min patterns

Analysis of the patterns was done using a self-written Matlab script. For all cases we analyzed the MinE signals, since the MinD signal gave less contrast and it makes no essential difference since both MinD and MinE signal the same qualitative and quantitative behavior of the patterns. First, for each movie, the field of view (FOV) was rotated automatically to make chambers fully vertical by calculating the standard deviation of each line along the average intensity of the movie and finding the angle at which this value is minimized. Next, the frames of the movies were thresholded using one of the built-in thresholding methods of Matlab and segmented automatically into chambers by finding intensity steps along the horizontal and vertical dimensions of an average or STD frame of the movie. We implemented a human-controlled correction step for both the rotation and segmentation steps in order to bypass program imperfection. The intensity of the Min proteins in each chamber was then recorded and the average intensity in each chamber was calculated separately and subtracted in order to produce separate movies for each chamber. The velocity of propagation was calculated by incorporating codes for the U-track 2.1.3 Matlab package that was developed in the Danuser lab (*Jaqaman et al., 2008*). For calculating the wavelength, a line, featuring the propagating direction was generated automatically based on the U-track results. Next, the intensity along this line was recorded for each frame of the background subtracted movie and a kymograph was generated. A peak finder Matlab routine (written by Jacob Kerssemakers) was used in order to locate the peaks distance along each line of the kymograph, and a histogram of the peak distances was generated. A Gaussian fit to the histogram was then used in order to find the mean value of the wavelength in each chamber. For calculating wavelengths on flat supported lipid bilayer (SLB), we used the same algorithm with the slight modification that the line along the waves' propagation direction was drawn manually. Velocities of the wave propagation on 2D flat SLB were calculated by rotating the peaks' kymograph and minimizing the average standard deviation along each line of the rotated kymograph. From the angle that minimizes the standard deviation and the error in the angle one can easily infer back the propagation velocity. Figures and graphs for this article were prepared using the programs: ImageJ (RRID:SCR_001935), SciDAVis (RRID:SCR_014643), Inkscape (RRID:SCR_014479) and GIMP (RRID:SCR_003182).

## Calculation of the protein concentration in the chambers

Our method of measuring the concentration of the Min proteins is based on the fact that, whereas MinD and MinE bind the membrane and thus one cannot know their concentration inside the microchambers a priori, the concentration of a cytosolic protein will always be equal to the concentration that is injected into the device. This means that the known concentration of a cytosolic fluorescent protein, such as GFP, can be used as a calibration tool in order to infer the actual concentration of the Min proteins in the microchambers by comparing the relative fluorescence, given that the relation between the relative fluorescence of GFP and either MinD or MinE is known for conditions where the Min proteins cannot bind the membrane. Thus, in order to measure the concentrations of the Min proteins in our chambers, we used a combination of a fluorescent measurement of the Min proteins signal in the chambers relative to their fluorescence at different concentrations in a cuvette measured with a fluorometer, and compared these values to the fluorescence of purified green fluorescence protein (deGFP - a variant of GFP see *Shin and Noireaux, 2012*), which was a kind gift of the Christophe Danelon lab ($\left[C_{GFP}^{Stock}\right] = 15 \pm 0.9 \mu M$ as measured by the Pierce 660 assay from Thermo Fisher scientific).

The concentration of a fluorescence species is related to the signal of a detector ($S$) according to the formula:

$$S = [C] \cdot B \cdot Qeff \cdot I \cdot \tau, \qquad (1)$$

where $[C]$ is the concentration of the fluorescent species, $B$ is the brightness of the species, $Qeff$ is the quantum efficiency of the detector at the emission wavelength of the fluorescent species, $I$ is the intensity of the light source at the excitation wavelength of the fluorescent species, and $\tau$ is the acquisition time of the detector. Upon defining $F \equiv Qeff \cdot I \cdot \tau$, we obtain $S_{Min} = F_{Min} \cdot B_{Min} \cdot [C_{Min}]$ for the Min proteins, and similarly $S_{GFP} = F_{GFP} \cdot B_{GFP} \cdot [C_{GFP}]$ for GFP. Since this equation can be written for both the bulk fluorometer ($S^{Flu}$) and for the microscopic imaging of the chambers ($S^{Mic}$) one obtains the following two equations:

$$\frac{S_{Min}^{Flu}}{S_{GFP}^{Flu}} = \frac{[C_{Min}^{Flu}] \cdot F_{Min}^{Flu} \cdot B_{Min}}{[C_{GFP}^{Flu}] \cdot F_{GFP}^{Flu} \cdot B_{GFP}}, \tag{2}$$

$$\frac{S_{Min}^{Mic}}{S_{GFP}^{Mic}} = \frac{[C_{Min}^{Mic}] \cdot F_{Min}^{Mic} \cdot B_{Min}}{[C_{GFP}^{Mic}] \cdot F_{GFP}^{Mic} \cdot B_{GFP}}, \tag{3}$$

where, for each variable, the superscripts $Mic$ or $Flu$ represent the case for the microchamber or the fluorometer cuvette, respectively. By defining $F^{Mic} \equiv \frac{F_{Min}^{Mic}}{F_{GFP}^{Mic}}$ and $F^{Flu} \equiv \frac{F_{Min}^{Flu}}{F_{GFP}^{Flu}}$ and rearranging *Equation 2 and 3*, one obtains:

$$[C_{GFP}^{Flu}] = \frac{S_{GFP}^{Flu}}{S_{Min}^{Flu}} \cdot F^{Flu} \cdot \frac{B_{Min}}{B_{GFP}} \cdot [C_{Min}^{Flu}], \tag{4}$$

$$[C_{GFP}^{Mic}] = \frac{S_{GFP}^{Mic}}{S_{Min}^{Mic}} \cdot F^{Mic} \cdot \frac{B_{Min}}{B_{GFP}} \cdot [C_{Min}^{Mic}]. \tag{5}$$

Since the GFP concentration is known and the same for the fluorometer and microchambers, $[C_{GFP}^{Mic}] = [C_{GFP}^{Flu}]$ and we can equate *Equation 4 and 5* to obtain:

$$[C_{Min}^{Mic}] = [C_{Min}^{Flu}] \cdot \left( \frac{S_{GFP}^{Flu}}{S_{Min}^{Flu}} \cdot \frac{F^{Flu}}{F^{Mic}} \cdot \frac{S_{Min}^{Mic}}{S_{GFP}^{Mic}} \right) \equiv [C_{Min}^{Flu}] \cdot \mathcal{F}. \tag{6}$$

Note that $F^{Mic}$ and $F^{Flu}$ are pure machine factors, they can be calculated from the known specs of the microscope camera and fluorometer detector, the relative intensities of the microscope and fluorometer light sources, and the acquisition times in both case.

To solve *Equation 6* and thus deduce the value of $[C_{Min}^{Mic}]$, one can adapt one out of two strategies that are mathematically equivalent. In the first approach, one substitutes all measured $S$ values to *Equation 6*, as well as $[C_{Min}^{Flu}] = [C_{Min}^{Syr}]$, where $[C_{Min}^{Syr}]$ is the concentration of the Min protein in the syringe that was used in order to inject the Min proteins into the microchambers, to thus obtain $[C_{Min}^{Mic}]$. In the second strategy, which was in fact applied in our work, we account explicitly for the possibly nonlinear dependence of fluorescence intensity versus concentration. Since for varying protein concentrations, $\mathcal{F}$ is a function of $S_{Min}^{Flu}$, one can convert the measured $S_{Min}^{Flu}$ values (see *Figure 5—figure supplement 1a,b*) to values for $\mathcal{F}$ and read off the value of $[C_{Min}^{Flu}]$ where $\mathcal{F}=1$. At this particular point we thus obtain the unknown Min concentration $[C_{Min}^{Mic}]$ in the microchambers. Note that for MinE, the calibration curve (*Figure 5—figure supplement 1b*) was linear while it was found to be nonlinear for MinD (*Figure 5—figure supplement 1a*). We do not fully understand the source of this non-linearity, but we suspect that it is related to some dequenching effect. However, the behavior was clearly reproducible as it was repeated in three independent experiments. Note also that the background signals in the microchambers in all fluorescence channels were measured before the injection of the proteins and were subtracted for the signal in the microchambers in the presence of the proteins. Similarly, the background values in the fluorometer were measured and subtracted from the protein signals.

## Observation of Min oscillation *in vivo*

Strain FW1919 (W3110 [minDE :: exobrs-sfGFP-minD minE-mKate2 :: frt]) (*Wu et al., 2016*) was used in order to view Min oscillations *in vivo*. Cells were inoculated into LB and were grown overnight at

37°C with 250 rpm shaking. In the morning, the cells were diluted 1:100 into M9 media + 0.2% glucose and continued to grow under the same conditions until they reached an OD of 0.3. Next, the cells were divided into two fractions. The first fraction was directly transferred to an M9 (+0.2% glucose) agar pad and were observed under the microscope to collect movies of Min oscillations in wild type cells. Cephalexin, (100 µg/ml, Sigma-Aldrich) was added to the second fraction and the cells continued to grow under the same conditions until they reached an OD of 0.94. Then, they were diluted 1:3 into fresh M9 +0.2% glucose and Cephalexin, placed on a similar agar pad, and were observed under the microscope to collect Min oscillations movies in filamentous cells. In all cases, movies were taken with the same Olympus IX81 setup that was used for the microchambers experiments. The temperature of the microscope was kept at 36.7°C and the acquisition was done using an Olympus UplanApo objective (100X, NA 1.35).

Movies were analyzed twice. First, to create *Supplementary file 1a–d* and the corresponding movie, the cells boundaries were identified by thresholding and the particle detection tool of ImageJ. Second, the midline of the cells was identified using a self-written script in Matlab based on thresholding and dissection of the cell boundaries into two halves (see *Caspi, 2014*). The midline was used in order to dissect the cells to separate areas each one with a width of 0.2 µm. The fluorescence in each area was summed and was used in order to construct the kymographs (*Supplementary file 1e–h*).

## Acknowledgement

This work was supported by the Netherlands Organization for Scientific Research (NWO/OCW), as part of the Frontiers of Nanoscience program, and the European Research Council for ERC Advanced Grant SynDiv (No. 669598). We would like to thank Petra Schwille for supplying the Min proteins strains, Christophe Danelon and Pauline van Nies for the kind gift of the GFP, Jacob Kerssemakers for help with the analysis algorithm, and Fabai Wu, Erwin Frey and Jacob Halatek for inspiring discussion.

## Additional information

### Funding

| Funder | Grant reference number | Author |
|---|---|---|
| Nederlandse Organisatie voor Wetenschappelijk Onderzoek | Frontiers of Nanoscience program | Yaron Caspi Cees Dekker |
| European Research Council | No. 669598 | Yaron Caspi Cees Dekker |

The funders had no role in study design, data collection and interpretation, or the decision to submit the work for publication.

### Author contributions

YC, Conception and design, Acquisition of data, Analysis and interpretation of data, Drafting or revising the article, Contributed unpublished essential data or reagents; CD, Conception and design, Analysis and interpretation of data, Drafting or revising the article

### Author ORCIDs

Yaron Caspi, http://orcid.org/0000-0003-0328-0186

## Additional files

### Supplementary files

• Supplementary file 1. Examples of *in vivo* oscillations in live cells. (a–d) Four examples of in vivo MinD oscillations in cells with various length of strain FW1919 (see *Wu et al., 2016*) Mol Sys Biol 12: 873). Panels show montages of the corresponding supporting *Supplementary file 2*. Time difference between frames is 12 s. Scale bar (5 µm) is shown to the left of panel (d). Cell boundaries are marked

in white. (a) Pole-to-pole oscillations in a wild type cell of normal length (3.8 μm). (b) A triple-node oscillations in a Cephalexin-division-inhibited cell with a length of 11.9 μm. (c) Multiple-node oscillations in a Cephalexin-division-inhibited cell (19.9 μm). (d) Aberrant triple-node oscillations in a Cephalexin-division-inhibited cell (14.4 μm). For the last case, instead of the regular pattern, where triple-point-oscillations occur between a middle zone and the two poles (see panels (b and f)), the middle zone seems to originate in the center of the cell and while traveling toward one pole, a MinD zone is established on the opposite pole. (e–h) Corresponding kymographs of the MinD intensity along the cells' length for panels (a–d) respectively. Scale bar for the kymographs are shown next to panel (h).

• Supplementary file 2. Examples of Min oscillations *in vivo* in strain BN1919 (see *Wu et al., 2016*) Mol Sys Biol 12: 873). Movie corresponds to *Supplementary file 1*. Fluorescence signal represents MinD oscillations.

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
