## [Decision Letter]

[Editors’ note: this article was originally rejected after discussions between the reviewers, but the authors were invited to resubmit after an appeal against the decision.]

Thank you for submitting your work entitled "Mapping out Min protein patterns in fully confined fluidic chambers" for consideration by *eLife*. Your article has been reviewed by three peer reviewers, and the evaluation has been overseen by a Reviewing Editor and Naama Barkai as the Senior Editor. The following individuals involved in review of your submission have agreed to reveal their identity: Yu-Ling Shih (Reviewer #2).

Our decision has been reached after consultation between the reviewers. Based on these discussions and the individual reviews below, we regret to inform you that your work will not be considered further for publication in *eLife*.

Although the reviewers find the 3D in vitro chambers interesting, there are still significant questions regarding the novelty of the findings, compared to previously published work (as pointed out by reviewer number 1). The extended exploration of parameter range, do not seem by itself sufficient to warrant publication in *eLife*. We hope that these reviews would help the authors to improve their manuscript.

*Reviewer #1:*

The authors describe a new microfluidic assay to study the self-organization of Min proteins in vitro, a protein system that has been extensively studied in the 15 years. I think the paper is a nice continuation of previous work, however the results presented have been published already in previous papers. Accordingly, it does not significantly contribute to our understanding of the Min system in particular or pattern formation in general.

The authors rightly note that the theoretical understanding of Min patterns in vivo is still incomplete and that in vitro observations can help to fill this gap. As correctly stated in their paper, this can be achieved by making the in vitro systems more and more similar to the in vivo situations. The authors have developed a microfluidic-based assay that allows to study Min protein patterns in three-dimensional membrane covered compartments. However, most observations have been previously made by other labs. For example, although the authors claim that they are the first to use "truly confined 3D structures", Zieske et al. 2013 and 2014 did fully confine the proteins in 3D (the proteins could not diffuse out of their compartments). It is therefore not really surprising that their findings basically match previous reports.

According to the authors, their main findings are:

1) Three different dynamical behavior dependent on the geometrical chamber parameters.

A number of previous studies have previously demonstrated the influence of geometry on Min protein pattern formation in vitro and it is now well established that aspect ratio and total size of the confining compartment can lead to either oscillations, running waves or more complex patterns (Schweizer et al. 2011, Zieske & Schwille 2013, 2014). Similar to Wu et al. (2015), the authors present a detailed statistical analysis, on which basis they claim to have discovered a new regime of "spiral rotations". However, one could argue that this is a mere although not very convincing reclassification of existing knowledge. It certainly does not contribute much to our understanding of Min protein patterns (see below).

2) Rotational behavior

According to the authors, their most striking observation is the rotational behavior of the Min waves that "occupies a large part of the phase diagram". However, it's well established that Min proteins, like many other reaction-diffusion systems, form spiral waves and that these spiral waves are more readily observed in large chambers with small aspect ratios is not really surprising (small chambers or *E. coli* cells could be just too small to accommodate a complete spiral. Accordingly, the patterns in small geometries simply represent "cropped" spirals). In fact, this observation was shown by two papers from the Schwille lab (Schweizer 2011, Zieske 2014). According to the authors their data suggests that the symmetry breaking mechanism in vivo could be different in vivo, but I do not understand how the data presented justifies this hypothesis.

3) Min protein wavelength

Caspi & Dekker have observed a decrease in wavelength depending on two factors, increase in viscosity, which has already been observed by Schweizer et al. 2011 and Martos et al. 2015, as well as an increase in temperature, which is a valid observation, but also not too surprising. They also suggest that confinement could lead to decrease in wavelength, most probably due to depletion of material, but a direct evidence backed up by additional controls is missing for this hypothesis. For example, one could quantify the ATP hydrolysis rate, membrane binding and detachment rates at different temperatures as well as the wavelength at different total protein concentrations. (One minor comment in this context: for the quantification of protein concentrations the significant autofluorescence of PDMS has not been taken into account, this could have lead to an overestimation of protein concentrations inside of the chambers. In addition, I am very confused about the nonlinear fit shown in Figure 5—figure supplement 1).

In summary, the data presented by Caspi & Dekker largely represents a duplication of previous work. Most observations are identical with previously published work and they not allow making any novel major conclusion about the Min protein system. However, I do think the development of the microfluidic assay is a great achievement and I recommend publication in a more specialized, e.g. method-focussed, journal than *eLife*.

*Reviewer #2:*

This article by Caspi and Dekker systematically analyzed the Min protein patterns in the confined 3D environment, highlighting importance of the geometrical cues in driving the pattern formation of the Min proteins in vitro. The authors conclude that the Min protein pattern is a function of the geometrical factor, which is determined by the microchamber dimension. The authors also examined effect of molecular crowding and temperature on the in vitro patterns and compared with the known patterns observed in *E. coli*. Although difference did exist, it was not sufficient to explain the scale difference between in vivo and in vitro observations. There are additional important points raised by the authors, including limitation for the current theoretical models to be capable of predicting all patterns, and the difficulty in reproducing the Min patterns in vitro at the same size scale as in vivo.

Few major comments:

1) While it is clear that the 'fully confined three dimensional chambers' were made, it is unclear on whether the supported lipid bilayer was coating the entire inner surface of the enclosed chamber so that the Min proteins could truly diffuse and react in a confined 3D space. The point is worth noting in order to distinguish from the experiment performed by Zieske and Schwille (2014; Figure 1). If the membrane coating was restricted to one surface or some areas of the chamber, please explain whether such discrepancy would or would not affect the conclusion.

2) It will be useful to compare shape, dimension, and aspect ratio (L/W) of the microchambers in this work with Zieske's design in the result or Discussion section.

3) For images, videos, and kymographs, it is unclear on which protein (MinD-Cy3 or MinE-Cy5 or merged images) is presented or analyzed.

4) The protein concentration measurement in the chamber is confusing, because the internal GFP control would remain in solution but the Min proteins would come on and off the membrane.

*Reviewer #3:*

The Min protein system is the best studied reaction-diffusion system that encodes intracellular spatial information. Although it appears simple, two proteins, a membrane and ATP, we are still a long way from understanding how this system interacts with the geometry of the cell to provide a spatial signal. In particular, there is a gulf between in vivo patterning data and artificial systems. Specifically, most artificial systems are not enclosed in a three-dimensional membrane system.

This paper provides extensive experimental data on how a three-dimensional enclosure affects the patterning produced by the Min proteins: MinD and MinE. The major result is to provide an "atlas" of patterning as a function of 3D confining chambers. Dynamic patterning is classified into: oscillations, travelling waves and spiral waves (rotations). The chamber geometry dictates the preference of the Min system for one of these classes of patterning. This work greatly expands our understanding of the link between patterning and chamber geometry. It comes closest to bridging the gap between in vivo observations and in vitro systems.

I have only minor criticisms of this work. The oscillations described in the manuscript and the attached videos are quite different from the Min oscillations observed in vivo. The spatial distribution of Min proteins is more akin to a one-dimensional travelling wave that originates in the centre of the chamber and moves to one pole at which time a new wave arises in the centre and translates to the opposite pole. The difference between this and in vivo observations (obvious when one compares the kymographs presented with published in vivo kymographs) may well be due to the size gap that still exists between these chambers (minimum dimensions 10 μm x 10 μm x 2.4 um) with bacterial cells. A clear comparison of the in vitro oscillation with in vivo oscillation should be presented and discussed.

[Editors’ note: what now follows is the decision letter after the authors submitted for further consideration.]

Thank you for resubmitting your work entitled "Mapping out Min protein patterns in fully confined fluidic chambers" for further consideration at *eLife*. Your revised article has been favorably evaluated by Naama Barkai (Senior Editor), a Reviewing Editor, and three reviewers.

The manuscript has been improved but there are some remaining issues that need to be addressed before acceptance, as outlined below:

After consultation with the Senior Editor, I decided to accept the recommendations of reviewers 2 and 3. In the revised version, please address the minor points raised by these reviewers, particularly with respect to proofreading to remove typos. There is no need to further address comments by reviewer 1.

*Reviewer #1:*

The authors set out to better understand the mechanism of Min protein pattern formation.

The main findings of this work according to the authors are:

1. Geometric confinement has an influence on the patterns formed

2. Increased crowding and temperature reduce the wavelength

3. Spirals are the underlying mechanism of symmetry breaking.

Regarding these findings, the following observations have been made previously in:

1) Influence of geometric confinement/depletion of proteins:

Schweizer, J., Loose, M., Bonny, M., Kruse, K., Mönch, I., & Schwille, P. (2012). PNAS, 109(38), 15283-15288.

Zieske, K., & Schwille, P. (2013). Angewandte Chemie 52(1), 459-462.

Zieske, K., & Schwille, P. (2014). *eLife*, 3. http://doi.org/10.7554/*eLife*.03949

Zieske, K., Schweizer, J., & Schwille, P. (2014). FEBS Lett, 588(15), 2545-2549.

Vecchiarelli, A. G., Li, M., Mizuuchi, M., Hwang, L. C., Seol, Y., Neuman, K. C., & Mizuuchi, K. (2016). PNAS, 113(11), E1479-88.

2) Influence of crowding has been studied in the following papers:

Schweizer, J., Loose, M., Bonny, M., Kruse, K., Mönch, I., & Schwille, P. (2012). PNAS, 109(38), 15283-15288.

Zieske, K., & Schwille, P. (2014). *eLife*, 3. http://doi.org/10.7554/*eLife*.03949

In addition, the following paper studied in detail the effect of other factors, such as salt concentration and membrane charge:

Vecchiarelli, A. G., Li, M., Mizuuchi, M., & Mizuuchi, K. (2014). Mol Microbiol, 93(3), 453-463.

So what is the main contribution of the paper from Caspi et al?

The have mapped out the influence of geometry in more detail, which motivated them to construct a phase diagram (Figure 3) and to conclude that:

3) "Not wavelike patterns, but spirals are the underlying mechanism of symmetry breaking" (see response letter).

I have no idea what this even means, "spirals" is a shape and *not* a mechanism, so what are the authors actually talking about here? The authors write that "This propensity to constitute different patterns suggests different symmetry breaking mechanisms. " So what are the different mechanisms? In what sense are they different? How do existing models fail to explain their findings? Does it help to understand what is going in the cell?

The authors make big claims, but in the end don't deliver much more than a rather descriptive paper. I do not see how their "findings" help to understand how Min proteins oscillate, i.e. how behavior on the molecular level gives rise to large scale patterns. When the authors write in their response that "Zieske and Schwille (2013) observed some patterns that vaguely resemble our spirals in their round chambers (Figure 4), but the authors did not dwell at all on the phenomena of spirals or spiral formation," it sounds to me much more like semantics than a scientific argument, just like their discussion about spirals, cropped spirals, and waves and the corresponding phase diagrams. I do not see what we have gained from this in my opinion rather arbitrary categorization of patterns that have been described before.

How does this paper contribute to the field or to science in general? The authors have developed a microfluidic system to study protein self-organization under controlled conditions. This is the major achievement of this study and this is what I think the authors should focus on.

Have the authors contributed significantly to our understanding of the mechanism of the Min waves? Does it go significantly beyond what was shown in a large number of previous publications in this and other journals? Do we now better understand how the symmetry breaks and what gives rise to spatial patterns? Do we better understand how the waves respond to confinement?

I honestly do not think so.

*Reviewer #2:*

The revised manuscript has addressed most questions and improved the clarity. The significance of the Min waves in the 3D confined space of difference dimension and aspect ratio is stated. I support its publication, with few comments to be addressed.

Figure 2, illustrations of the quadrant scheme. The single-frame and STD illustrations for oscillations (b) do not match with the MinE data. The highest concentration of MinE is found at the trailing edge of a wave. The single-frame images of traveling waves (c) and rotations (d) have the same problems. The concentration gradient or the arrow indicating the direction of wave propagation need to be corrected.

Discussion paragraph 7: The new images provided for comparison between in vivo and in vitro oscillations brings in new confusions. This manuscript reports the in vitro behavior of MinE, but additional data of MinD oscillation in vivo are presented for comparison. It becomes necessary to describe the different features of MinE and MinD oscillations in vivo to justify the authors' points. However, since the discussion is rather long already, I am not sure about the necessity of having this part in the discussion.

Discussion paragraph eight-: I am uncomfortable about 'in vivo traveling waves', because it is an abnormal phenomenon caused by abnormal function of MinE. Unless emphasizing the 'in vivo traveling waves' could give new physiological insights, it is not necessary to overinterpret the in vitro waves. The in vitro study on its own has provided rich information from physical perspectives.

*Reviewer #3:*

I believe that the authors have clearly (and emphatically) addressed all issues raised in the original three reviews.

I am pleased that the authors now include Figure 1—figure supplement 1 showing the oscillations in live cells. It is clear from these images, the kymographs and [Supplementary-material SD2-data] that the in vivo patterns differ from the in vitro patterns. Clearly in vivo Min patterning is more complex and will require further research to gain a complete understanding. Nonetheless, Caspi & Dekker's present work makes a substantial contribution to our understanding of this amazing dynamical system.

---

## [Author Response]

[Editors’ note: the author responses to the first round of peer review follow.]

*Although the reviewers find the 3D* in vitro *chambers interesting, there are still significant questions regarding the novelty of the findings, compared to previously published work (as pointed out by reviewer number 1). The extended exploration of parameter range, do not seem by itself sufficient to warrant publication in eLife. We hope that these reviews would help the authors to improve their manuscript.*

*Reviewer #1:*

The authors describe a new microfluidic assay to study the self-organization of Min proteins in vitro, a protein system that has been extensively studied in the 15 years. I think the paper is a nice continuation of previous work, however the results presented have been published already in previous papers. Accordingly, it does not significantly contribute to our understanding of the Min system in particular or pattern formation in general.

We strongly disagree with Referee 1 on these last two points, that also are factually incorrect – see the detailed discussion below.

We nevertheless thank the referee for his/her critical remarks of our work. Although we do not agree with his/her main criticism, it has surely helped us to rewrite the conclusion section in a manner that will better clarify the innovation of our work. In particular, we now stress the difference between our work and previous work more strongly in the discussion.

*The authors rightly note that the theoretical understanding of Min patterns in vivo is still incomplete and that in vitro observations can help to fill this gap. As correctly stated in their paper, this can be achieved by making the in vitro systems more and more similar to the in vivo situations. The authors have developed a microfluidic-based assay that allows to study Min protein patterns in three-dimensional membrane covered compartments. However, most observations have been previously made by other labs. For example, although the authors claim that they are the first to use "truly confined 3D structures", Zieske et al. 2013 and 2014 did fully confine the proteins in 3D (the proteins could not diffuse out of their compartments). It is therefore not really surprising that their findings basically match previous reports.*

Zieske et al. worked with ‘bathtub-shaped’ compartments that were only partly (at their bottoms) coated with lipid membrane, and where the aqueous solvent was evaporating quickly. We now better describe the advantages of our approach in comparison to the work from Zieske et al.,.

Next to that, we like to point out that our data very significantly go beyond the results published by Zieske et al., as we explore a greatly expanded range of geometrical parameters. Only in the limited regime (of narrow structures) that Zieske et al. explored, we reproduced the patterns that they observed (which is a good thing). We therefore find it unfair to conclude that it is “not really surprising that their findings basically match previous reports”.

*According to the authors, their main findings are:*

1) Three different dynamical behavior dependent on the geometrical chamber parameters.

*A number of previous studies have previously demonstrated the influence of geometry on Min protein pattern formation in vitro and it is now well established that aspect ratio and total size of the confining compartment can lead to either oscillations, running waves or more complex patterns (Schweizer et al. 2011, Zieske & Schwille 2013, 2014). Similar to Wu et al. (2015), the authors present a detailed statistical analysis, on which basis they claim to have discovered a new regime of "spiral rotations". However, one could argue that this is a mere although not very convincing reclassification of existing knowledge. It certainly does not contribute much to our understanding of Min protein patterns (see below).*

*2) Rotational behavior*

*According to the authors, their most striking observation is the rotational behavior of the Min waves that "occupies a large part of the phase diagram". However, it's well established that Min proteins, like many other reaction-diffusion systems, form spiral waves and that these spiral waves are more readily observed in large chambers with small aspect ratios is not really surprising (small chambers or E. coli cells could be just too small to accommodate a complete spiral. Accordingly, the patterns in small geometries simply represent "cropped" spirals). In fact, this observation was shown by two papers from the Schwille lab (Schweizer 2011, Zieske 2014).*

Referee 1 criticizes the novelty and importance of our observation that rotational waves underlie the establishment of oscillations. Apparently we did not clarify this well enough in the discussion, but we note that we agree with referee 1 that the oscillations observed in in vitro studies represent a form of cropped spirals. In fact, this is exactly the key point that we were trying to make in this paper. Indeed, this point is essential for an understanding of the symmetry breaking and pattern formation mechanisms of the Min system in vitro. Unfortunately, the referee did not appreciate the importance of this point.

Furthermore, contrary to the claims of referee 1, this point was not made in previous work such as the papers that the referee refers to (Schweizer et al. (2011) or Zieske and Schwille (2013, 2014)). To be concrete, we here cite several relevant passages from these articles which we know very well:

I) Zieske and Schwille (2013) observed some patterns that vaguely resemble our spirals in their round chambers (Figure 4), but the authors did not dwell at all on the phenomena of spirals or spiral formation. On the contrary: for example, the authors write: “our findings provide evidence that wavelike patterns indeed underlie the same mechanism as the oscillatory pattern…”. Thus, the main claim of the article was that wavelike patterns, and not spirals, are the underlying mechanisms of the symmetry breaking in the in vitro Min system – in stark contrast to our key finding.

II) In Zieske and Schwille (2014), the authors concentrated on structures that are 10 μm or smaller. They only briefly studied structures with larger width and summarized their findings with these words: “In compartments with a small width of 10 μm, only oscillations along the long axis were supported. (…) Even larger widths resulted in more complex oscillation patterns.” Similar passages exist at page 13 and in the caption of video 4 of this article. Thus, where we foundorder in the form of spirals, Zieske and Schwille did not find any order at all, but merely alluded to ‘more complex oscillation patterns’. This shows the significance of our approach and the novelty of the observations for the symmetry breaking mechanism of the Min system.

III) Finely, Schweizer et al. (2012) did not study spirals at all. Instead, they were interested in the way that Min waves adapt to geometrical cues. Thus, also in this article the importance of spirals was not acknowledged.

We have now integrated a more detailed discussion of the importance of cropped spiral for the understanding of in vitro waves into the Discussion section.

*According to the authors their data suggests that the symmetry breaking mechanism in vivo could be different in vivo, but I do not understand how the data presented justifies this hypothesis.*

This notion is based on the importance of the cropped spiral for the formation of oscillatory pole-to-pole behavior, combined with the lack of such observations in vivo. As Referee 1 suggested in point 2, it may indeed be true that *E. coli* cells are just too small to accommodate spirals. However, when cell were widened more than twice as large as the typical length scale in vivo, spirals were still not observed, but instead transversal or higher-order longitudinal oscillations were seen (Wu et al., Nature Nanotechnology, 2015). This stands in direct contrast to the in vitro behavior that is reported in the current manuscript. This propensity to constitute different patterns suggests different symmetry breaking mechanisms.

*3) Min protein wavelength*

*Caspi & Dekker have observed a decrease in wavelength depending on two factors, increase in viscosity, which has already been observed by Schweizer et al. 2011 and Martos et al. 2015, as well as an increase in temperature, which is a valid observation, but also not too surprising.*

We disagree that our study merely reproduces previously published findings. Numerically, the reduction in velocity for the viscous 2D case was much smaller (at maximum a factor of 3.5 – Martos et al. 2015 figure S7) than what was observed by us (a factor of 10). This can be attributed to the crucial difference between our setup (confined 3D) and the conditions under which the effect of viscosity was studied before (unconfined 2D surfaces). Indeed, recent theoretical work by Frey and co-workers suggests that it is probably linked to the depletion zone in the 3D-confined structures, and thus an intrinsic effect related to confinement that is qualitatively different from the previous studies that studied Min proteins in an unconstrained hemisphere above a surface.

Furthermore, we note that our study provides the first time that the influence of temperature on the Min system has been studied in vitro and that a correspondence (in velocity enhancement) and difference (in wavelength reduction) between the in vitro and in vivo behavior is discussed. Thus, also regarding this case, we find it unfair to suggest that our finding are a mere reproduction of previous results. Assessing this finding as ‘unsurprising’ only because some aspects of the behavior were similarly observed in vivo (while, in fact, other properties were not) is unreasonable, since this logic diminishes every in vitro study that reproduces some aspects of the in vivo behavior. In the words of this referee him/herself: “The authors rightly note that the theoretical understanding of Min patterns in vivo is still incomplete and that in vitro observations can help to fill this gap. As correctly stated in their paper, this can be achieved by making the in vitro systems more and more similar to the in vivo situations.” Indeed, And for this reason it is actually very interesting to study the temperature dependence and discuss the differences and similarities between the in vitro and in vivo studies.

Finally, we thank the referee for the reference to Martos et al., regarding the effect of viscosity on the Min waves. We now added a reference to this paper and compared their results regarding the wavelength and the velocity of the waves to ours, see Results and Discussion sections.

They also suggest that confinement could lead to decrease in wavelength, most probably due to depletion of material, but a direct evidence backed up by additional controls is missing for this hypothesis. For example, one could quantify the ATP hydrolysis rate, membrane binding and detachment rates at different temperatures as well as the wavelength at different total protein concentrations.

We do want to emphasize the importance of our observation of a reduced wavelength in confinement, as this is a key point that we were trying to make. To our understanding, the true confinement of the Min system in 3D with a small height (5 times smaller than achieved in the work of Zieske and Schwille) restricts the volume accessible for Min protein to exchange between the bulk and membrane-bound forms, and doing so results in the reduced length scale. This observation is at the heart of studying the effect of confinement on the Min system in vitro. As was very recently shown theoretically (but not yet tested experimentally), the formation of a depletion zone as result of the combination between diffusion and geometry is a key parameter in the formation of intracellular gradients (Thalmeier et al. PNAS 113, 2016). This theoretical prediction is novel, while, to the best of our knowledge, no in vitro evidence exists yet in the literature to verify this important novel concept. Our finding of a reduced wavelength in the microchambers, just because of the confinement, is therefore a novel experimental finding that significantly expands the scientific understanding of intracellular protein gradients as well as their reconstitution in vitro.

Referee 1 suggests that we should add additional controls for backing up our conclusion that 3D confinement is related to a reduced wavelength. In the confined chambers, our observed reduced wavelength corresponds well with the difference in the diffusion rates at the two temperatures and thus it is natural to interpret it, as we did, in the context of differences in the diffusion rate. On 2D surfaces, by contrast, we observed a reduced wavelength that cannot be understood from diffusion rate differences alone. In that case, the effect indeed can result, as referee 1 suggests, from the difference in various reaction parameters. However, the behavior on 2D surface is not the focus of our manuscript which is already loaded with data on different conditions in the chambers. We therefore feel that it is appropriate to leave any extension on this observation for future work. This is emphasized by the fact that no one knows how to accurately measure reaction rates such as the MinD attachment rate or the cooperativity rate, and thus to disentangle the effect of different reaction parameters on this phenomenon.

*One minor comment in this context: for the quantification of protein concentrations the significant autofluorescence of PDMS has not been taken into account, this could have lead to an overestimation of protein concentrations inside of the chambers.*

This was accounted for. As described in the manuscript, we measured the background fluorescence for the three channels (GFP, Cy3 and Cy5) before the proteins were injected and subtracting these values from the values measured in the presence of GFP, MinD-Cy3 and MinE-Cy5. This therefore corrects for any background contributions, including that to autofluorescence of PDMS. In order to make it clearer, we now stated more explicitly in the Supplementary Information that the values inside the chambers represent the measured intensity minus the background.

*In addition, I am very confused about the nonlinear fit shown in Figure 5—figure supplement 1.*

We agree with the referee that this behavior is surprising. Indeed, we were surprised ourselves at first and had to convince ourselves that this is not an artifact. Therefore, we reproduced the measurement independently with three different samples on three different days to ensure that there was no mistake. The reported behavior is what is observed, an experimental fact, see subsection “Calculation of the protein concentration in the chambers”.

*In summary, the data presented by Caspi & Dekker largely represents a duplication of previous work. Most observations are identical with previously published work and they not allow making any novel major conclusion about the Min protein system. However, I do think the development of the microfluidic assay is a great achievement and I recommend publication in a more specialized, e.g. method-focussed, journal than eLife.*

It is nice to hear that the referee praises our development of the microfluidic assay as a great achievement. But more importantly, as documented above, we think that it is unwarranted criticism of referee 1 to suggest that our findings are a mere repetition of previously published work about the Min system. In contrast, we feel that our finding finally uncovers a number of essential points regarding the Min system that were previously unnoticed and that fundamentally expand the scientific understanding of this important model system. For this reason, we think (as also expressed clearly by Referee 3) that our results are very important and merit publication in a high-profile general venue for publication, viz., *eLife*.

*Reviewer #2:*

*[…] Few major comments:*

*1) While it is clear that the 'fully confined three dimensional chambers' were made, it is unclear on whether the supported lipid bilayer was coating the entire inner surface of the enclosed chamber so that the Min proteins could truly diffuse and react in a confined 3D space. The point is worth noting in order to distinguish from the experiment performed by Zieske and Schwille (2014; Figure 1). If the membrane coating was restricted to one surface or some areas of the chamber, please explain whether such discrepancy would or would not affect the conclusion.*

We thank the referee for this remark. Apparently this was not clear enough. It is indeed very important to stress that in our case, all walls of the microchambers were coated with a supported lipid bilayer (SLB). To further clarify this, we added a few sentences to the Results section. We also added a remark about this in the Discussion section.

2) It will be useful to compare shape, dimension, and aspect ratio (L/W) of the microchambers in this work with Zieske's design in the result or Discussion section.

We agree. A paragraph that describes the differences in geometry between our case and that of Zieske et al. was now added to the Discussion section.

*3) For images, videos, and kymographs, it is unclear on which protein (MinD-Cy3 or MinE-Cy5 or merged images) is presented or analyzed.*

In all cases we have analyzed the MinE signal, since the MinD signal gave less contrast. This makes no essential difference, since both MinD and MinE signal the same qualitative and quantitative behavior of the patterns. We have now added a note about this in the figure captions (Figure 1–Figure 4 and Figure 6) as well as in the relevant supplementary figures and in the Materials and methods.

*4) The protein concentration measurement in the chamber is confusing, because the internal GFP control would remain in solution but the Min proteins would come on and off the membrane.*

The latter is exactly the point. Indeed, it is hard to infer the Min concentration directly due to the fact that the Min proteins bind to membrane-covered surface of our microchambers. For this reason, we deviced a method to measure the protein concentration that is indeed based on the fact that, while the Min proteins bind the membrane, cytocolic proteins like GFP will have a well-known concentration in the microchambers which can be taken as a reference. We added a paragraph to the Supplementary Information that explains the logic behind the protein concentration measurement.

*Reviewer #3:*

*[…] I have only minor criticisms of this work. The oscillations described in the manuscript and the attached videos are quite different from the Min oscillations observed in vivo. The spatial distribution of Min proteins is more akin to a one-dimensional travelling wave that originates in the centre of the chamber and moves to one pole at which time a new wave arises in the centre and translates to the opposite pole. The difference between this and* in vivo *observations (obvious when one compares the kymographs presented with published* in vivo *kymographs) may well be due to the size gap that still exists between these chambers (minimum dimensions 10 μm x 10 μm x 2.4 um) with bacterial cells. A clear comparison of the in vitro oscillation with in vivo oscillation should be presented and discussed.*

We thank the referee for this useful suggestion. We have carried out additional supporting experiments on Min oscillations in normal and filamentous *E. coli* cells to address this. A new figure, Figure 1—figure supplement 1 has been added to the supplementary information as well as a corresponding movie ([Supplementary-material SD2-data]). A detailed explanation of the method that was used to construct these files has been added to the supplementary information. A discussion of the in vivo oscillations in comparison to the in vitro ones has been added to the Discussion section of the main text.

[Editors’ note: the author responses to the re-review follow.]

*[…] Reviewer #2:*

*The revised manuscript has addressed most questions and improved the clarity. The significance of the Min waves in the 3D confined space of difference dimension and aspect ratio is stated. I support its publication, with few comments to be addressed.*

*Figure 2, illustrations of the quadrant scheme. The single-frame and STD illustrations for oscillations (b) do not match with the MinE data. The highest concentration of MinE is found at the trailing edge of a wave. The single-frame images of traveling waves (c) and rotations (d) have the same problems. The concentration gradient or the arrow indicating the direction of wave propagation need to be corrected.*

We would like to thank the referee for her sharp eye. Indeed, this comment is entirely justified regarding the single-frame illustrations. In response, we have now changed (i) the direction of the color gradient in the single-frame illustration for oscillations; (ii) the representation of the wave intensity in the single-frame illustration so that it more closely represents the dynamics of the Min waves (where the intensity of MinE falls off slower on the side of the wave that is further from its direction of propagation); (iii) The direction of the arrow for the spiral rotation so that the spiral propagates from the highest MinE intensity to the lowest. (iv) The corresponding single frame representation of the oscillations and spiral rotations so that they more closely match the experimental data.

The referee’s criticism is however not justified for the STD illustrations, as can for example be seen by comparing the illustrations to the actual data (Figure 2) in the 20x10 microns case (for oscillations), 30x30 microns (for spiral rotations), and 60x50 microns (for waves). We therefore have left these images as is.

*Discussion paragraph 7: The new images provided for comparison between in vivo and in vitro oscillations brings in new confusions. This manuscript reports the in vitro behavior of MinE, but additional data of MinD oscillation in vivo are presented for comparison. It becomes necessary to describe the different features of MinE and MinD oscillations in vivo to justify the authors' points. However, since the discussion is rather long already, I am not sure about the necessity of having this part in the discussion.*

We added these new data upon the request of referee 3 (and he/she writes that he/she is very pleased that we did so, see below). We are aware that this extends the Discussion section slightly, but we feel (following referee 3) that it adds an important perspective.

The referee is correct regarding the fact that the data is presented for MinE in vitro and MinD in vivo. As we will argue now, however, this difference is unimportant because – while specifics are different for MinD versus MinE – the major pattern characteristics discussed in this paper are the same:

Usually in vivo MinE forms an intensive band at the trailing zone of the oscillation cup (the E-ring) and thus, its kymograph fingerprints are somewhat different from those of MinD, This is what the referee comments on. Usually, such an in vivo MinE fluorescent signal is obtained by an ectopic expression of a recombinant fluorescent MinE from an inducible promoter, which typically leads to overexpression of MinE. In the strain that we used, however, the recombinant fluorescent MinE was expressed from its native locus on the chromosome under the native Min promoter (Wu et al. Molec. Syst. Biol. (2016) 12: 873). This is nice since it expresses the MinE at its natural level but this also led to only a weak fluorescence MinE signal, making it very hard to obtain clear MinE images in vivo (which is why we resorted to reporting the MinD signal). Note that in this strain MinD and MinE proteins do move concurrently and no E-ring is observed (see for example Results and Video EV3 of the Supplementary Information of that paper by Wu et al.). Thus, for this endogenous-expression strain, the MinE and MinD kymographs are similar. Interestingly, in vitro, MinD and MinE move concurrently as well. Reports of in vitro waves showed no evidence for an E-ring structure, and while the intensity MinE wave profile was slightly different from that of MinD (Loose et al. Science (2008) 320, 789-792; Loose at al. Nat Struct Mol Biol (2011), 18, 577-583) the MinE again moved concurrently with MinD. Summing up, similar patterns were observed for MinD and MinE and the fact that we report MinE signal in vitro and MinD signal in vivo does not make any difference for the conclusions that are drawn here.

For clarification, we added a few sentences to the Discussion section.

*Discussion paragraph eight-: I am uncomfortable about 'in vivo traveling waves', because it is an abnormal phenomenon caused by abnormal function of MinE. Unless emphasizing the 'in vivo traveling waves' could give new physiological insights, it is not necessary to overinterpret the in vitro waves. The in vitro study on its own has provided rich information from physical perspectives.*

We entirely agree with the referee that in vivo traveling waves represent a form of abnormal behavior which occurs if the MinE function is impaired or in other cases as explained in the main text. That was exactly the point we wanted to make: in contrast to the in vitro behavior, the in vivo traveling waves that have been reported are an abnormal phenomenon.

In order to clarify this, we now added a sentence in the Discussion section.

*Reviewer #3:*

*I believe that the authors have clearly (and emphatically) addressed all issues raised in the original three reviews.*

*I am pleased that the authors now include Figure 1—figure supplement 1 showing the oscillations in live cells. It is clear from these images, the kymographs and [Supplementary-material SD2-data] that the in vivo patterns differ from the in vitro patterns. Clearly in vivo Min patterning is more complex and will require further research to gain a complete understanding. Nonetheless, Caspi & Dekker's present work makes a substantial contribution to our understanding of this amazing dynamical system.*

We would like to thank again referee 3 for his/her recommendations and for the improvement to this manuscript as results of his/her comments.